# An approach to MOFaxanes by threading ultralong polymers through metal–organic framework microcrystals

Tomoya Iizuka[1], Hiroyuki Sano[2], Benjamin Le Ouay [1,3], Nobuhiko Hosono [2] ✉ & Takashi Uemura [2] ✉

Mechanically interlocked architecture has inspired the fabrication of numerous molecular systems, such as rotaxanes, catenanes, molecular knots, and their polymeric analogues. However, to date, the studies in this field have only focused on the molecular-scale integrity and topology of its unique penetrating structure. Thus, the topological material design of such architectures has not been fully explored from the nano- to the macroscopic scale. Here, we propose a supramolecular interlocked system, MOFaxane, comprised of long chain molecules penetrating a microcrystal of metal–organic framework (MOF). In this study, we describe the synthesis of polypseudoMOFaxane that is one of the MOFaxane family. This has a polythreaded structure in which multiple polymer chains thread a single MOF microcrystal, forming a topological network in the bulk state. The topological crosslinking architecture is obtained by simply mixing polymers and MOFs, and displays characteristics distinct from those of conventional polyrotaxane materials, including suppression of unthreading reactions.

Rotaxane is a well-known family of interlocked molecules that consist of two major elements: a macrocyclic ring like a rotor and a threaded backbone string like an axle with bulky end caps[1,2]. Since rotaxanes were first proposed in 1961[3], several rotaxane structures comprising a variety of organic and inorganic elements have been synthesized[4–6]. Polyrotaxane, discovered in 1992[7,8], is a polythreaded version of rotaxane with numerous rings on a single polymer chain. This new class of mechanically interlocked molecules promoted the development of supramolecular soft materials exhibiting unique mechanical properties and functions[9–12]. However, thus far, previous studies on (poly)rotaxanes focused on the design of molecular-sized elements and were primarily discussed in the realm of supramolecular chemistry. Therefore, this characteristic feature of interlocked systems has not been extended to the design of novel solid materials from a nano- to a macroscopic standpoint.

Metal–organic frameworks (MOFs) are crystalline materials that have nanometer-sized pores resulting from the coordination of organic ligands to metal ions[13,14]. Since their discovery in the 1990s, MOFs have been extensively studied for separation, storage, catalysis, and transportation applications. Most of the previous studies, however, focused on small molecules as the targeted guests to be handled in the pores[15–17]. Hence, large polymers have rarely been considered guest molecules for MOFs. In general, the insertion of polymers into a nanospace is considered an unfavorable process because the entangled chains may need to uncoil, thereby resulting in a large entropic disadvantage. However, this notion has been discredited by recent demonstrations of the spontaneous infiltration of synthetic polymers and biomacromolecules, such as polyethylene oxide (PEO)[18,19] and DNA[20], respectively, into nanoporous MOFs, which resulted in a restrained conformation of the polymer chains in the nanopores[18,19,21].

[1]Department of Advanced Materials Science, Graduate School of Frontier Sciences, The University of Tokyo, 5-1-5 Kashiwanoha, Kashiwa 277-8561 Chiba, Japan. [2]Department of Applied Chemistry, Graduate School of Engineering, The University of Tokyo, Bunkyo-ku 113-8656 Tokyo, Japan. [3]Present address: Department of Chemistry, Graduate School of Science, Kyushu University, 744 Motooka, Nishi-ku, 819-0395 Fukuoka, Japan. ✉e-mail: nhosono@g.ecc.u-tokyo.ac.jp; uemurat@g.ecc.u-tokyo.ac.jp

However, most of these previous examples have used polymer chains with molecular weights (MW) in the range 600–20,000 g mol$^{-1}$ (contour length, $L_c$ = 5–150 nm for PEO as an example), whereas the size of the host MOF crystals was extremely large (typically, tens to hundreds of micrometers in diameter) compared to that of the guest polymers.

In this context, we envisioned a reversal of the conventional size relationship between a MOF and its guest, i.e., the use of ultralong guest polymers whose length significantly exceeds the size of the MOF particles. Here, we thread MOF microcrystals with ultralong polymer chains, thereby proposing a class of interlocked system, MOFaxane, that utilizes MOFs as the rotors on the polymer chain axles (Fig. 1a). In analogy with rotaxanes, we can define subclasses of MOFaxane, which include pseudoMOFaxane and polymeric analogs (Fig. 1). In this study, we synthesized polypseudoMOFaxane as the first example of the MOFaxane family, which opens up new avenues to construct numerous interlocked materials using MOF crystals as the functional rotary elements.

## Results

We used a nanoporous pillared-layer-type MOF microcrystal, $[Cu_2(bdc)_2(bpy)]_n$ (**1**, bdc = 1,4-benzenedicarboxylate, bpy = 4,4′-bipyridyl), as the host for an ultralong PEO guest[22,23]. Atomic force microscopy studies showed that the microcrystals of MOF **1** have a plate-like shape with a mean diameter and thickness of 640 nm and 80 nm, respectively (Fig. 2a). Powder X-ray diffraction (PXRD) of **1** showed typical peak broadening, reflecting the sub-micron crystal size (Supplementary Fig. 1). MOF **1** has a twofold interpenetrated structure and exhibits a dynamic structural change, i.e., the gate-opening behavior from a closed pore (**1**-*cp*) to an open pore (**1**-*op*) phase, upon inclusion of guest molecules in the nanopores[22] (Fig. 2b). The **1**-*cp* phase has narrow and unconnected pores, whereas the **1**-*op* phase has open voids with clearly defined pores in the interstices between the $[Cu_2(bdc)_2]_n$ layers (i.e., in the *ab* plane) passing along the [110] and [1−10] directions (Fig. 2b), corresponding to the short axis of the crystal. The distinct crystal structures of both the closed and open phases allowed us to monitor the guest occupation in **1** by PXRD measurements (Supplementary Fig. 1).

During our initial attempt, we inserted a relatively short PEO with an MW of 2000 g mol$^{-1}$ (**PEO2k**, $L_c$ = ~15 nm) into **1**. A mixture of **1**-*cp* and **PEO2k** powders (mixing ratio, **1**/PEO = 1/2, wt/wt) was heated at 120 °C and annealed for 90 min under a nitrogen atmosphere. At this temperature, molten PEO infiltrated **1**, forming a composite, **1/PEO2k$_{1/2}$** (see "Methods"). At the **1**/PEO weight ratio of 1/2, the amount of PEO was approximately 9-fold higher than the guest capacity of **1**-*op* that was previously determined for small guest molecules such as *N*,*N*-dimethylformamide (DMF) (0.23 g/g) (Supplementary Fig. 5)[22]. PXRD patterns of the mixture showed a remarkable structural change from the **1**-*cp* to the **1**-*op* phase, indicating the insertion of **PEO2k** into the nanopores of MOF **1** (Fig. 3a). We investigated the thermodynamic

origin of PEO insertion into **1** and calculated the adsorption enthalpy to be 32 kJ/mol (per repeating unit of PEO), which is larger than that measured for PEO adsorption into a typical pillared-layer-type MOF (See Supplementary Information and Supplementary Figs. 6–8)[21]. To thread all the microcrystals of **1** on the polymer chain, we considered ultralong PEO and **PEO4M** (MW = 4,000,000 g mol$^{-1}$, $L_c$ = ~30 μm), which is 2000 times longer than **PEO2k** and >300 times thicker than that of the particles of **1**, as potential guests (Supplementary Fig. 3). However, the structure change of **1** was barely observed for the ultralong **PEO4M** upon heating at 120 °C for 90 min unlike the instant infiltration of **PEO2k** (Fig. 3a). To investigate the infiltration process in detail, time-resolved PXRD measurement was performed for the mixtures of **1** and PEO at 120 °C. The fraction of **1**-*op* was calculated by the peak deconvolution analyses on time-resolved PXRD data using the representative diffractions of the **1**-*cp* and **1**-*op* phases, (1–10) and (200) at $2\theta$ = 16.9° and 16.4°, respectively (Fig. 3b). The results for **1/PEO2k** mixture revealed that the gate-opening process for **PEO2k** completed within ~5 min, indicating rapid infiltration of the PEO chains into **1**. We also performed the time-resolved PXRD measurements for PEOs with different MWs, **PEO10k** (MW = 10,000 g mol$^{-1}$), **PEO20k** (MW = 20,000 g mol$^{-1}$), **PEO200k** (MW = 200,000 g mol$^{-1}$), and **PEO4M**, under identical conditions. For **PEO10k** and **PEO20k**, the *cp*-to-*op* structure change happened but was slower than that observed for **PEO2k** (Fig. 3b). Interestingly, for **PEO200k** ($L_c$ = ~1.5 μm), whose chain length is >5 times longer than that of **1** particle, the gate-opening of **1** still took place although the rate was significantly reduced as the fraction of **1**-*op* reached less than 20% even after 90 min. Eventually, the infiltration of **PEO4M** was observed to be too slow to induce an appreciable gate opening of **1** at this condition. This clear correlation between the MW of PEOs and the gate-opening rate indicates that the diffusion of PEO chains in the pores can be a predominant regulation factor of this infiltration process. As a control experiment, we examined the insertion of PEO (MW = 20,000 g mol$^{-1}$) end-capped with bulky *tert*-butyldiphenylsilyl (TBDPS) groups (**PEO20k**-TBDPS), whose projection diameter is larger than the window size of **1**-*op*. Although the MW was in the insertable range, the end-capped PEO did not induce any structural change for **1** (Supplementary Fig. 9). This observation clearly shows that the spontaneous insertion of PEO chains always takes place from their termini.

Due to their harshly limited diffusion, it was considered difficult to directly introduce such ultralong PEO chains into **1** by molten-phase insertion. Surprisingly, however, instant insertion of **PEO4M** was achieved when the insertion experiment was performed in the presence of a solvent, chloroform, as a so-called sacrificial guest. The mixture of **1** and **PEO4M** in chloroform (**1**/PEO ratio of 1/0.30, wt/wt) was heated at 80 °C to remove the solvent gradually, then completely dried under the reduced pressure (0.3 kPa) at 100 °C (see "Methods" for detailed experimental procedures). The resulting material, denoted as **1/PEO4M$_{1/0.3}$**, showed a full opening of **1** to give a **1**-*op* phase in the PXRD

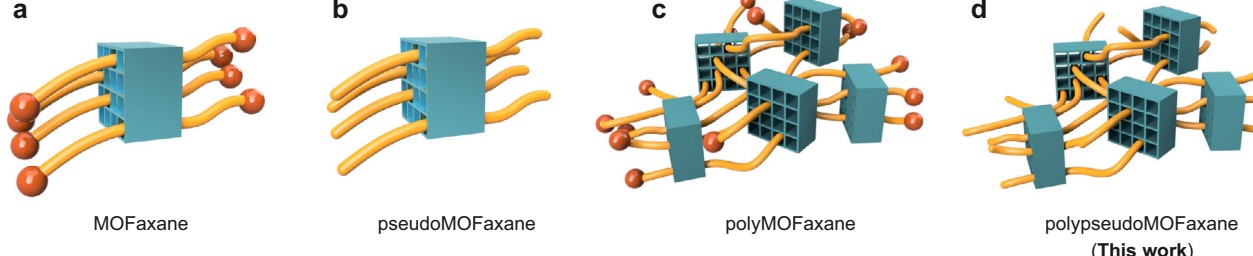

**Fig. 1 | Definitions and graphical representations of the MOFaxane family.** **a** MOFaxane, **b** pseudoMOFaxane, **c** polyMOFaxane, and **d** polypseudoMOFaxane. PseudoMOFaxane has no end cap on the polymer chains, while MOFaxane has end caps by definition. PolyMOFaxane and polypseudoMOFaxane have network structures in which each polymer chain threads multiple MOF microcrystals, forming a topological network. In this work, polypseudoMOFaxane, **d** was fabricated.

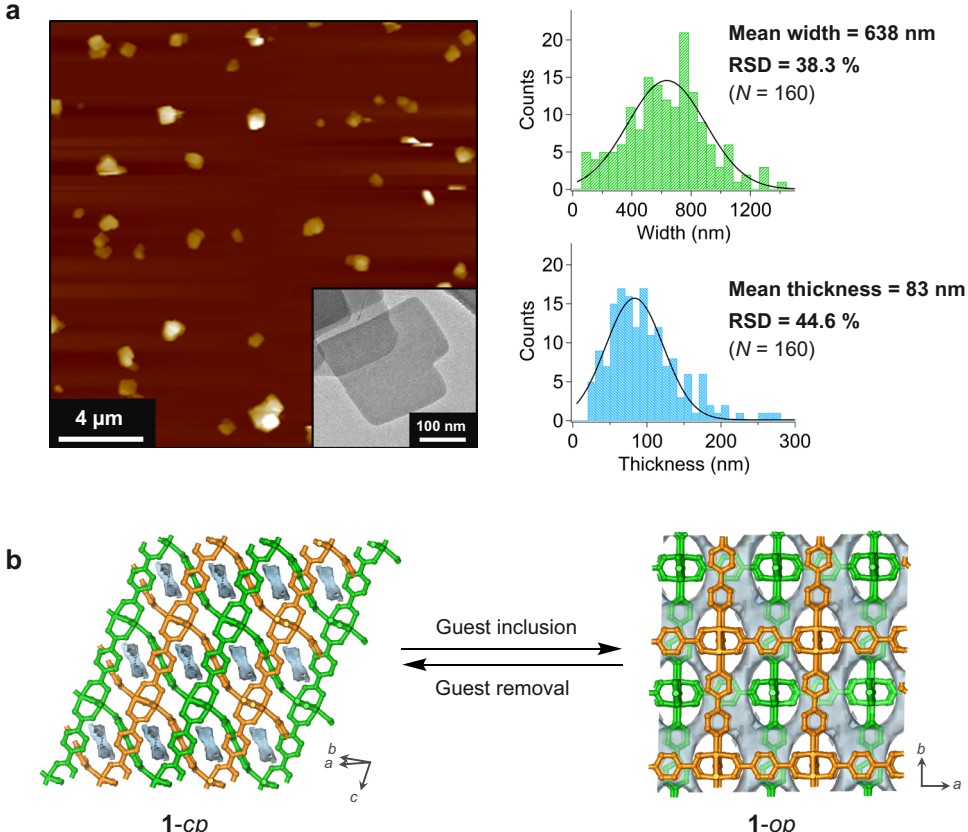

**Fig. 2 | Structures of the flexible pillared-layer-type MOF [Cu₂(bdc)₂(bpy)]ₙ (1).**
**a** Atomic force microscopic (AFM) and transmission electron microscopic (TEM) images of the synthesized **1** (left and left inset, respectively) and width and thickness distribution of the particles of **1** using measurements from five AFM images (right). **b** Crystal structure and the surface of the crystallographic voids of the **1**-closed pore (*cp*) (top) and **1**-open pore (*op*) (bottom) phases (View is along the direction perpendicular to the bpy ligand). Voids were probed using spheres of 2.4 Å diameter.

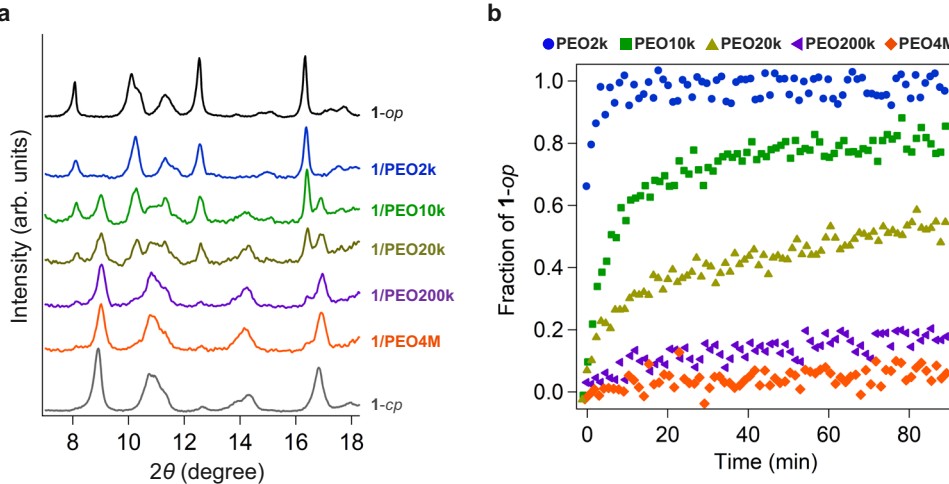

**Fig. 3 | Structural changes in 1 induced by PEO infiltration. a** Powder X-ray diffraction (PXRD) profiles for the mixture of **1/PEO2k** (blue), **1/PEO10k** (green), **1/PEO20k** (yellow), **1/PEO200k** (purple), and **1/PEO4M** (orange) (mixing ratio, **1/**PEO = 1/2, wt/wt) after the heating at 120 °C for 90 min under nitrogen atmosphere. Black and gray lines denote the PXRD patterns of **1**, including methanol as the guest data (Fig. 4a), suggesting the infiltration of **PEO4M** into **1**. We consider molecule, showing the **1**-*op* phase, and **1** in the evacuated state, showing the **1**-*cp* phase, respectively. **b** Time evolution plots of the **1**-*op* fraction of in contact with molten PEO with MW of 2000 (**PEO2k**) (blue), 10,000 (**PEO10k**) (green), 20,000 (**PEO20k**) (yellow), 200,000 (**PEO200k**) (purple) and 4,000,000 (**PEO4M**) (orange), monitored in situ at 120 °C under nitrogen atmosphere.

that the **PEO4M** insertion was greatly promoted because the presence of solvent molecules preabsorbed in **1** opened the gate before insertion was initiated. A similar phenomenon where one component opens the gate to simultaneously introduce another component in the pore is commonly observed in mixed-gas separation experiments for flexible MOFs[24,25]. In the present study, this behavior, which is otherwise controversial in practical gas separation applications of flexible MOFs, is an important advantage in facilitating polymer insertion. Additionally, the presence of solvent molecules may also contribute to the mobility of

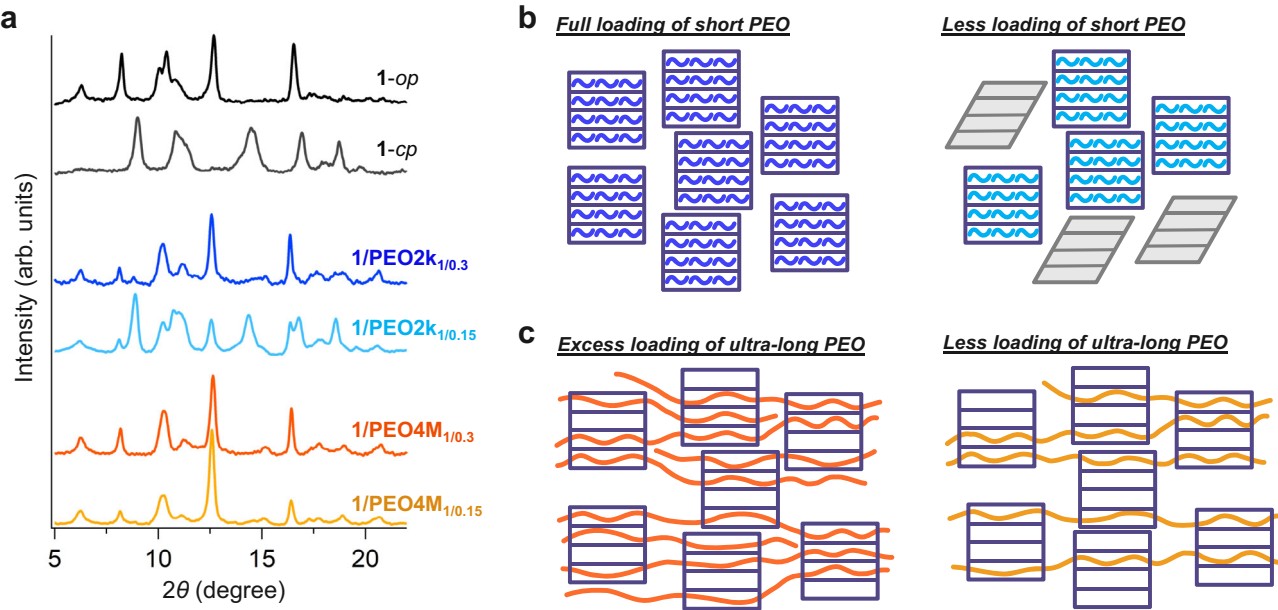

**Fig. 4 | Structural changes of 1 depending on the feed amount of PEO. a**, PXRD patterns for **1/PEO2k$_{1/0.3}$** (blue), **1/PEO2k$_{1/0.15}$** (light blue), **1/PEO4M$_{1/0.3}$** (orange), and **1/PEO4M$_{1/0.15}$** (yellow). **b, c**, Schematic of the infiltration of short PEO (**b**) and ultralong PEO (**c**) into MOF **1** using PEO amounts below and above the maximum loading capacity of **1**. When short PEO chains with an amount less than the maximum loading capacity are used (**b**, bottom), a limited number of particles of **1** (purple squares) convert into the *op* phase and quickly incorporate PEO (curved

lines), whereas the remaining MOF particles that may have had less contact with the PEO chains remain in the *cp* phase. In contrast, ultralong PEO, which is much longer than the particle size of **1**, can penetrate and bridge across multiple crystals of **1** simultaneously. This polythreading structure leads to the complete opening of MOF **1** even though the amount of PEO used is less than the theoretical maximum (**c**, bottom).

chain termini searching for the pore entrance at the onset of the insertion event, countering diffusion limitation.

Considering the extremely long length of **PEO4M**, which largely exceeds the particle size of MOF **1**, we assume that a single strand of **PEO4M** penetrates through multiple microcrystals of **1**. This potentially affords the polypseudoMOFaxane structure (Fig. 1). To support this idea, we investigated the scenario in which the feed amount of PEO is less than the ideal maximum capacity of **1**. In the preliminary test, we inserted the short chain **PEO2k** into **1** using a **1**/PEO ratio of 1/0.15 (wt/wt), (**1/PEO2k$_{1/0.15}$**). A comparison of the PXRD patterns of **1/PEO2k$_{1/0.15}$** and **1/PEO2k$_{1/0.3}$** shows the difference in the **1**-*op* fraction (Fig. 4a). **1/PEO2k$_{1/0.3}$** demonstrated a full phase shift to **1**-*op*, whereas the **1**-*cp* phase of **1/PEO2k$_{1/0.15}$** co-existed with the **1**-*op* phase, resulting in a mixture and indicating that a portion of **1** is still left closed. As each **PEO2k** chain is sufficiently short to be entirely encapsulated in the crystal of MOF **1**, the **1**-*op* phase of the MOF collected all the free **PEO2k** molecules, leaving the excess microcrystals in the closed phase (Fig. 4b). Such coexistence of open and closed phases is commonly observed for flexible MOFs during the adsorption of gaseous molecules[26,27]. In contrast, the adsorption of ultralong polymer chains has different consequences. As observed in the PXRD data for **1/PEO4M$_{1/0.15}$**, the theoretically insufficient feed of **PEO4M** afforded a fully open phase (Fig. 4a). Interestingly, we found that even 1/0.05 (**1/PEO4M$_{1/0.05}$**) showed a high fraction of **1**-*op* phase (81.5%, Supplementary Fig. 10). This can be attributed to the formation of a polythreading configuration, in which a few PEO chains can induce several particles of **1** to open simultaneously via multiple cycles of penetration (Fig. 4c). It was previously demonstrated that the introduction of a small quantity of a polymeric guest in a flexible MOF induces the open phase in the host, imparting an accessible porosity to the otherwise nonporous frameworks in their original closed phase[28]. Therefore, we concluded that the polythreading configuration is the most likely microstructure pattern of the **1/PEO4M** composites, as evidenced by the fully transformed PXRD pattern for **1/PEO4M$_{1/0.15}$**.

In analogy with the conventional cyclodextrin-based polyrotaxane, we estimated the percent coverage of the axial PEO with **1** microcrystal. Based on the actual pore occupancy evaluated by gas adsorption analysis, the averaged coverage is calculated to be 51% for **1/PEO4M$_{1/0.3}$** composite (see Supplementary Information and Supplementary Fig. 11). As deduced from the coverage value, free PEO chains are likely to present in between threaded microcrystals even for the high MOF loading composite. The bridging PEO chains tether multiple microcrystals together to form loose agglomerates in the solution phase. Indeed, the formation of such agglomerated structures was observed in the laser-scattering particle size distribution measurements (Supplementary Fig. 12) and AFM images of the dispersed samples (Fig. 5 and Supplementary Figs. 13 and 14).

To investigate the effect of the MOFaxane structure on the physical properties of the composite, the crystallization of PEO was investigated using DSC upon cooling from 100 to 0 °C at various cooling rates (Supplementary Fig. 15). Based on the DSC cooling profiles, the PEO crystallization rates were determined for the composites **1/PEO2k$_{1/1}$** and **1/PEO4M$_{1/1}$** and compared with those of the pristine PEOs, **PEO2k** and **PEO4M**, respectively. Interestingly, a significant reduction of the crystallization rate was observed for **1/PEO4M$_{1/1}$**, whereas the crystallization rate of **1/PEO2k$_{1/1}$** was comparable to **PEO2k** (Supplementary Fig. 16). For a more quantitative analysis, we constructed Friedman plots to determine the effective activation energy of crystallization (Supplementary Fig. 17, Supplementary Table 1). The effective activation energies at various stages of crystallization are shown in Fig. 6a. In the initial stage of crystallization, the activation energy decreases as the crystallization proceeds for all PEOs and composites. This suggests that the crystallization process is promoted by the progressive formation of its own nucleation centers in the PEO domain. However, only **1/PEO4M$_{1/1}$** demonstrates a significant increase in its activation energy as crystallization progressed, which is the opposite trend to the other samples, the pristine PEOs (**PEO2k** and **PEO4M**) and **1/PEO2k$_{1/1}$**. This result can also be attributed to the

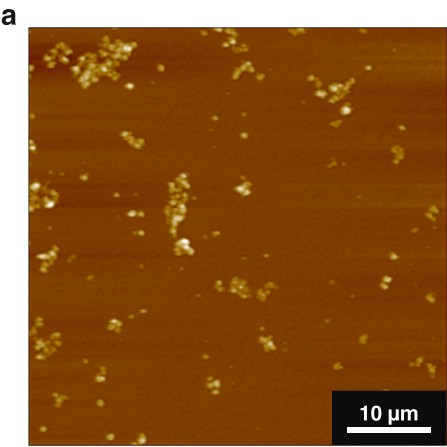

**Fig. 5 | Morphologies of polypseudoMOFaxane. a, b** AFM images of **1/PEO4M$_{1/1}$** composite deposited on a mica substrate. The **1/PEO4M$_{1/1}$** composite was dispersed by stirring for 5 min in chloroform (1 mg/mL) and deposited on a mica substrate by spin coating (2500 rpm, 5 s) at 25 °C. **a**, 50 μm × 50 μm topographic image (scale bar, 10 μm). **b** 20 μm × 20 μm topographic image (scale bar, 4 μm).

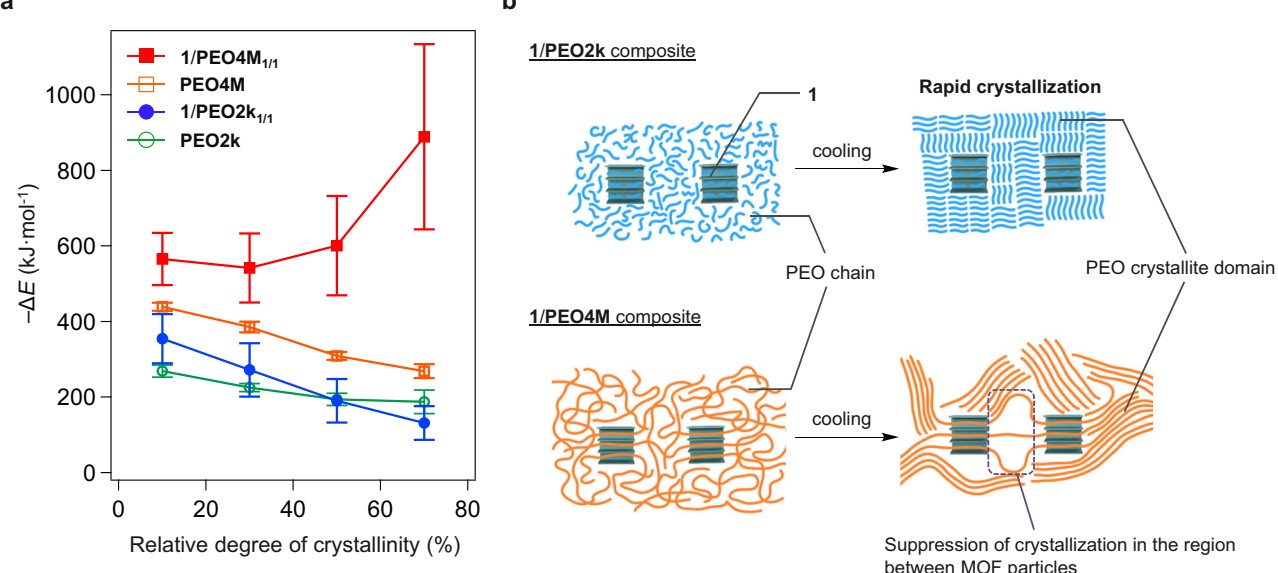

**Fig. 6 | Effect of the polyMOFaxane configuration on the properties of the composite. a** Friedman plot of the effective activation energies for the crystallization of PEO chains for **PEO2k** (green), **PEO4M** (orange), **1/PEO2k$_{1/1}$** (blue), and **1/PEO4M$_{1/1}$** (red). Data are presented as mean ± SD (*N* = 3). **b** Schematic of the suppression of PEO crystallization in the MOF/PEO composites. The PEO chains of **1/PEO2k$_{1/1}$** (top) at the inter-particle region of **1** crystallize normally. In the **1/PEO4M$_{1/1}$** composite, PEO chains are tethered by the crystals of **1** owing to the polythreadisng structure, which suppresses crystallization.

formation of the polyMOFaxane structure in **1/PEO4M$_{1/1}$**. For the composite with a short PEO chain, **1/PEO2k$_{1/1}$**, the chains existing outside the crystal of MOF **1** are sufficiently mobile to crystallize upon cooling, similar to the pristine samples (Fig. 6b). In contrast, for the composite with the polypseudoMOFaxane configuration **1/PEO4M$_{1/1}$**, the mobility of the PEO chains was restricted by the bridging structure. Thus, the crystallization of the PEO chains in the inter-particle region of **1** was effectively hindered (Fig. 6b). This indicates that the formation of the polyMOFaxane structure has a significant effect on the thermal properties of axle polymers owing to its distinct penetrating structure.

Although the axle PEO has no bulky endcaps, the unthreading reaction was significantly suppressed for the fully loaded composite **1/PEO4M$_{1/1}$**, even after rigorous washing of the composite for 6 h using a good solvent of PEO, dichloromethane (DCM) (See Methods). The DCM-washed **1/PEO4M$_{1/1}$** did not recover its **1**-*cp* form, whereas a control experiment using a composite with the short PEO, **1/PEO2k$_{1/1}$**,

showed **1**-*cp* formation in PXRD analysis after the DCM-washing treatment (Supplementary Fig. 20). $^1$H NMR analysis of **1/PEO4M$_{1/1}$** digested in a D$_2$O/EDTA-4Na mixture showed that ~25% of the feed **PEO4M** remained even after washing (Supplementary Figs. 21 and 22). Notably, the observed residual amount of **PEO4M** is close to the full capacity (0.23 g/g) of MOF **1**. In contrast, a significant loss of the PEO guest was found for **1/PEO2k$_{1/1}$**, leaving only ~3% PEO of the initial feed (Supplementary Figs. 21 and 23). Interestingly, the washing experiment on **1/PEO4M$_{1/0.3}$**, in which the inter/intra-molecular entanglements of PEO chains are expected to be lower than that of **1/PEO4M$_{1/1}$** due to the higher coverage value, also showed a large loss of **PEO4M** (see Supplementary Information and Supplementary Fig. 21). These results suggest that chain entanglements play a significant role in preventing the unthreading reaction. Therefore, it can be considered that the present polypseudoMOFaxane structure is mainly stabilized by kinetic factors that rely on the entanglements of ultra-long PEO chains penetrating the MOF particles.

To investigate the influence of the MOFaxane structure on the bulk mechanical properties of the composite, thick films of the poly-threading composite **1/PEO200k$_{1/20}$** (**1**/PEO = 1/20, wt/wt) and **1/PEO200k$_{1/10}$** (**1**/PEO = 1/10, wt/wt) were prepared via a slurry casting method and subjected to uniaxial tensile tests at 25 °C and 70 °C (Supplementary Figs. 24 and 25). As a result, an increase in the elastic modulus was observed for both composite films. In particular, the **1/PEO200k$_{1/10}$** film showed a 1.5-fold (25 °C) and a 1.2-fold (70 °C) enhancement of the elastic modulus in comparison to a reference specimen comprised of pristine **PEO200k**, which underpins the anticipated networking structure (Supplementary Tables 2 and 3). It should be noted that the shorter PEO, **PEO2k**, did not give such a robust and free-standing film. As conventional polyrotaxane-based materials show many characteristic mechanical features[10,11], which are ascribed to the dynamic motion of cyclodextrin rings sliding on the axle chain, we envisioned that a similar effect could be implemented in the polyMOFaxane networks. Interestingly, the tensile tests conducted at a higher temperature of 77 °C revealed a reversed trend in the elastic modulus, where the composite film was less elastic compared to the pristine film (Supplementary Fig. 25 and Supplementary Table 3). This temperature dependence suggests that the contribution of possible filler effect is ruled out, and it may reflect the sliding motion that gets promoted at this temperature. However, at this point, the sliding motion of MOF crosslinks on the PEO axle could not be further char-acterized by the present method due to technical limitations caused by the significantly slow diffusion of the PEO chain in the pore (See Sup-plementary Information). Tuning of pore size and pore environments that dictate polymer–MOF interactions will allow for rational optimi-zation of the chain diffusion rate and, therefore, bulk mechanical properties. To this end, further MOF engineering, as well as studying the thermodynamic rationale of this ultralong polymer insertion, are now being pursued in our group. Following this polymer insertion-based fabrication process, the polyMOFaxane networks are obtained by simply mixing polymers and MOF microcrystals, whereas the con-ventional topological polymer networks, e.g., polyrotaxane gels, need a crosslinking reaction to couple cyclodextrin rotors in their synthesis[9]. By taking full use of the simple fabrication procedures as well as the highly tunable feature of MOF structures, it should be possible to synthesize even more diverse MOFaxane architectures that will be counted as a family of mechanically interlocked materials[29–31].

## Discussion

MOF microcrystals were threaded with ultralong polymer chains to obtain a class of organic-inorganic hybrid materials possessing the interlocked rotaxane architecture, which we name MOFaxanes. The MOFaxane structure was formed via the gate-opening behavior of the flexible MOF host. A MOFaxane has a polythreaded structure in the bulk state, i.e., multiple polymer chains are threaded into a single MOF microcrystal to create a topologically crosslinked network structure. Tuning the pore size and pore structures that dictate polymer-MOF interactions and improving the penetration method will afford more diverse MOFaxane architectures and properties. We expect that MOFaxanes will promote the design of organic-inorganic hybrid materials that can be extended from the nano- to the macroscopic scale by extending previously reported concepts.

## Methods
### Materials
All reagents and chemicals used in this study were obtained from FUJIFILM Wako Pure Chemicals and Tokyo Chemical Industry unless otherwise noted. Deuterated solvents were purchased from Cam-bridge Isotope Laboratories. PEOs with MW of 2000, 10,000, 20,000, 200,000, and 4,000,000 g/mol, denoted as **PEO2k**, **PEO10k**, **PEO20k**, **PEO200k**, and **PEO4M**, respectively, were purchased from FUJIFILM Wako Pure Chemicals and Merk KGaA/Sigma-Aldrich, and

used without further purification. PEO was pulverized into fine pow-dery form prior to use.

### General instrumentations
$^1$H nuclear magnetic resonance (NMR) spectra were recorded using a Bruker AVANCE III HD spectrometer operating at 500 MHz with a PABBO probe. PXRD data was recorded on a Rigaku model SmartLab X-ray diffractometer using Cu Kα radiation. Analytical differential scanning calorimetry (DSC) was performed on a HITACHI High Tech model DSC-6020 and DSC-7020 differential scanning calorimeter. Thermogravimetric (TG) analysis was carried out on a Rigaku model ThermoPlus TG8120 TG analyzer. Transmission electron microscopic (TEM) image was observed at room temperature using a JEOL model JEM-2100 transmission electron microscope. N$_2$ adsorption measure-ments were performed on a MicrotracBEL model BELSORP II mini, and N$_2$ gas of high purity (99.9999%) was used. Samples were activated at 100 °C for 8 h before the measurement. Particle size distribution ana-lysis was performed using a HORIBA model LA-950 laser-scattering particle size distribution analyzer using CHCl$_3$ as the dispersion medium.

### Synthesis of [Cu$_2$(bdc)$_2$(bpy)]$_n$ (1)
**1** was synthesized using a modified version of the previously reported procedure[22]. To a methanol solution (320 mL) of H$_2$bdc (336 mg, 2.0 mmol), a methanol solution (20 mL) of AcOH (5.8 mL, 0.10 mmol) and Cu(OAc)$_2$·H$_2$O (400 mg, 2.2 mmol) was added at 25 °C. The mix-ture was left to stand for 3 days at 25 °C. Then, a methanol solution (100 mL) of bpy (156 mg, 1.0 mmol) was added to this mixture, which was left to stand for 2 days at 25 °C. A pale green precipitate was formed, which was collected by centrifugation (43,380×*g*, 15 min) and washed with DMF (25 mL) and methanol (25 mL) by decantation to afford **1** in the open form (**1**-*op*). Prior to use in the insertion experi-ments, **1** was activated under vacuum at 130 °C for 16 h, which afforded the guest-free closed form of **1** (**1**-*cp*) (562 mg, 92% yield based on bpy).

### Procedure for the direct insertion of molten PEO into 1
Direct insertion of molten PEO into **1** was performed on an XRD-DSC module attached to a Rigaku model Smart Lab X-ray diffractometer. A mixture of **1** and powdery PEO (**1**/PEO = 1/2, wt/wt, 4 mg in total) was placed on the aluminum sample plate (Rigaku). The temperature was controlled by using the XRD–DSC module with the following ramp and hold program: 25 °C | 10 °C/min (ramp) → 120 °C (target temperature) | 90 min (hold). To prevent undesirable oxidation and decomposition of PEO, 2,6-di-tert-butyl-4-methylphenol (3 wt%) was added to PEO, and PXRD measurements were performed under nitrogen flow (120 mL/ min). While heating to the target temperature, PEO melted. Upon heating, the PXRD patterns were recorded at regular time intervals. The time evolution plots of the fraction of **1**-*op* displayed in Fig. 3b were generated based on the height ratio of the two representative diffraction peaks of the **1**-*cp* and **1**-*op* phases, (1–10) and (200) at 2θ = 16.9° and 16.4°, respectively, for each time-resolved PXRD pattern.

### Procedure for the fabrication of 1/PEO composite
A chloroform suspension (5 mL) of **1** (30 mg) and PEO (30 mg) were mixed in a glass vial with vigorous stirring. The mixture was then heated at 80 °C in the glass vial without the cap for 3 h to slowly eva-porate the solvent. The mixture was further heated under vacuum at 100 °C for 3 h to completely remove the solvent and for the annealing in which PEO is inserted into **1**. In the case of **1/PEO4M**, PEO was preliminarily dissolved in chloroform (12 mL) and then mixed with the suspension of **1** in order to have a homogeneous mixture.

### Synthesis of PEO20k-TBDPS
**PEG20k** (0.60 g, 30 μmol) and imidazole (0.82 g) were dissolved in dehydrated DMF (20 mL). To the mixture was added

*tert*-butyldiphenylchlorosilane (TBDPS-Cl) (1.6 mL), and the mixture was stirred for 24 h at 50 °C. The reaction mixture was poured into ethanol (excess) and cooled at −10 °C. The formed white precipitate was collected by centrifugation and decantation. The product was successively dissolved in dichloromethane (1 mL) and reprecipitated from cold ethanol (excess, −10 °C). The reprecipitation procedure was repeated three times. The product was dried under vacuum at 50 °C for 10 h, affording **PEO20k**-TBDPS (0.57 g, 93% yield) as a white solid. The terminal conversion was calculated to be ~93% based on [1]H NMR data. [1]H NMR (500 MHz, CDCl$_3$): *d* (ppm) 1.04 (s), 3.64 (broad), 7.35–7.42 (m), 7.68 (m).

### Tensile tests of the composite film

Specimens used for uniaxial tensile tests were prepared by slurry casting from a chloroform suspension (10 mL) of **PEO200k** (362 mg) and guest-free **1**-*cp* (36.2 mg). The suspension was poured into a self-made PTFE round-shape mold (50 mm diameter), and the solvent was slowly evaporated in the air at 80 °C. The resulting thick film was further dried at 100 °C under a reduced pressure for 3 h. The film was punched out into proper shape and dimensions. For tensile tests at 25 °C, six dog-bone specimens (2 mm width, 100 μm thickness) were prepared and measured using a Shimadzu model AG-X plus universal tester with a 5 N local cell with the elongation rate of 20 mm/min to obtain the averaged value of elastic modulus, stress at yield, stress at fracture and fracture strains. For the tensile tests at 70 and 77 °C, rectangular specimens (5 mm width, 100 μm thickness) were prepared, and the elastic moduli were measured using a tensile test module attached to a TA Instruments model Discovery HR 10 rheometer with the elongation rate of 30 μm/s. The measurements were performed under temperature control in the environmental test chamber.

### Procedure for the solvent washing of the composites

A suspension of **1**/PEO composite (10 mg) in DCM (2.5 mL) was stirred at 25 °C. After a given time, the composite was collected by centrifugation (24,400×*g*, 15 min), rinsed with DCM (2.5 mL), and vacuum-dried at 50 °C for 6 h. After the washing process, the amount of PEO remaining in **1** was quantified by [1]H NMR using D$_2$O/EDTA-4Na mixture (0.05 M) as the solvent. The insoluble bpy ligand that precipitated out was removed by filtration before the NMR measurement.

## Data availability

All the data supporting the findings of this study are available in the article and its Supplementary materials and from the corresponding authors, N.H. and T.U., upon request.

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

## Acknowledgements

This work was supported by a KAKENHI Grant-in-Aid for Japan Society for the Promotion of Science (JSPS) Fellows (JP21J12230) (T.I.), JSPS KAKENHI Grant-in-Aid for Scientific Research (B) (JP21H01981) (N.H.), Scientific Research on Innovative Areas "Molecular Engine" (JP21H00385) (N.H.), and Scientific Research (A) (JP21H04687) (T.U.). N.H. acknowledge the financial support of the UTEC-UTokyo FSI Research Grant Program. T.U. is grateful for Data Creation and Utilization-Type Material Research and Development Project (JPMXP1122714694). TEM observation was performed using facilities of the Institute for Solid State Physics, the University of Tokyo.

## Author contributions

Conceptualization and supervision: N.H. and T.U.; Experiment and analysis: T.I., H.S., B.L., and N.H.; Writing: T.I., H.S., N.H., and T.U.

## Competing interests

The authors declare no competing interests.
