## [Peer Review File · Nature Communications]

An approach to MOFaxanes by threading ultralong polymers through metal–organic framework microcrystalsReviewers' Comments:

Reviewer #1:

Remarks to the Author:

The paper presents a simple approach to fabricate a so-called mechanically interlocked material, MOFaxane, in which the ultralong polymer chains thread microcrystals of metal-organic frameworks. Unlike most of the examples previously reported, the length of ultralong guest polymers here significantly exceeds the size of the MOF particles. It is worth noting that the cp-op structure change monitored with PXRD experiments proves the successful insertion of PEO chains into the nanopores of MOF and thus the authors thought the polythreaded structure is similar to the polyrotaxanes. Subsequently, the effects of the MOFaxane structure on the organic-inorganic hybrid materials' physical properties and bulk mechanical properties were investigated through a series of methods. Partial results are interesting and the project is worthy of investigation. However, there are several unreasonable explanations and more importantly, the proposed MOFaxane structure is not precise.

Some discussions may be helpful for the improvement of the manuscript before it is submitted again:

- (1) The authors attempted to name the MOF composite by learning from the linear polyrotaxane. A rotaxane contains at least three elements: macrocycle, axle, stoppers. In the MOF composite, there are no stoppers. Therefore, strictly speaking, such analogy is not suitable. In fact, there are polypseudorotaxanes which are similar to the MOF composite in structure.
- (2) As for the polypseudorotaxanes made of cyclodextrins (CDs) and PEOs, the covering ratio can be calculated so that we know how many CDs threaded into the backbone of PEOs. Similarly, it is meaningful to estimate how many MOFs thread into the backbone of PEO4M. Moreover, how much is the driving force for the threading event and how is the influence of threading numbers of MOF on the threading efficacy. In other word, when the first MOF is threaded on the backbone of PEO, will it accelerate or slow down the second one for threading?
- (3) The Figure 4c and Figure 5b are totally different on the description of the possible structure of the so called MOFaxane. What is the true structure? According to Figure 4c, it seems a linear structure, but for Figure 5b, it becomes a crosslinked network by using the MOF as a crosslinking point. From my point of view, I prefer to think the MOFaxane is a network and the numbers of threaded MOFs are not very high.
- (4) For the macroscopic mechanical characterization of the MOFaxane materials, the variations are too small to be utilized to explain the threading effect. As we know, solid additive can well reinforce the mechanical performance of polymers. Therefore, such minor changes in the parameters of mechanical properties can not be used directly. Moreover, the authors referred to the CDs-based sliding materials to explain their observation. If thing goes according to the author' hypothesis, the elastomeric modulus of the MOFaxane should decrease rather than increase a little bit. More suitable explanations and more measurements on the mechanical properties should be made for better understanding of the MOFaxane materials.

Some technical concerns for consideration:

- (1) Fig. 2b shows the gate-opening behavior from 1-cp to 1-op phase corresponding to the peaks of PXRD at 16.9° and 16.4°, respectively. Could you please plot the corresponding crystal forms [1-10] and [200] at Fig. 2b to make it a little bit more intuitive?
- (2) At the bottom of page 5, 'heated at 120 °C and annealed for 90 min under nitrogen atmosphere' is not mentioned in the Methods. Please check it and fill in the missing pieces.
- (3) The successful insertion of PEO4M was achieved in the presence of a sacrificial guest. For further verifying PEO4M's infiltration and exploring the internal structure, could you please evaluate the pore size distribution of the closed-MOF, MOF-CHCl₃ and MOF-PEO4M with nitrogen sorption experiment?
- (4) For 1/PEO2k, the co-existence of 1-op and 1-cp phase was observed by regulating the proportion of MOF and polymer chains. In order to gain insight into structure and minimize internal tangles, it's necessary to screen the ratio to find the critical point of PEO4M's content, below which the co-existence of 1-op and 1-cp phase in 1/PEO4M will occur.
- (5) PXRD alone is not a powerful method to prove the rotaxane-like conformation in the MOFaxane

demonstrated in Fig. 1. It is also possible that the conformation of MOF_{axane} is a tightly packed spherical structure due to strong chain entanglement effect and the interactions between MOF particles. Please use some more intuitive characterization methods to prove the rotaxane-like conformation.

(6) Compared with PEO_{2k}, what're the reasons for the obvious decrease of 1/PEO_{2k}'s effective activation energy after 50% crystallinity shown in Fig. 5a?

(7) Please discuss the internal mechanism by which the effective activation energy of 1/PEO_{4M} decreases first and then increases.

(8) In the unthreading experiment, we don't know if it was chain entanglement or multiple interactions between 1 and PEO_{4M} that prevented 1/PEO_{4M}'s unthreading reaction. Let's come back to question (4), the unthreading experiment should be taken under the circumstances that the internal tangles are minimized.

(9) The dynamic motion of MOF particles sliding on the axle chain may bring the material toughness enhancement. Please calculate and compare the toughness of several films according to their typical stress-strain curves shown in Supplementary Fig. 11 to explore the possible sliding motion.

Reviewer #2:

Remarks to the Author:

The authors reports threading the PEO into the MOF crystal and performed detailed study on the thermal degradation and the rate of crystallization of the polymer-MOF composite (1/PEO_x). This work is significant as the field is lacking in the formation mechanism in the polymer threaded MOF.

1) In terms of the structure and morphology of the polymer-MOF composite, SEM and or TEM images of the pristine MOF, and after loading PEO (1/PEO_{2k1/1} and 1/PEO_{4M1/1}), as the major results are for these two composites. Thus, more information on the structure and morphology on the composites.

2) From the synthesis, PEO is inserted in MOF by molten phase insertion. Could the authors explain how the polymer thread into the MOF pore? Detail discussion should be included in the maintext, in addition to Fig. 5b.

Also additional characterization like cross-section TEM maybe needed to prove PEO is threaded into MOF.

At the moment, it seems to be either the MOF are filling the interspace between the PEO polymer chain, or the bulk PEO polymer is wrapping the MOF, or coating the MOF. Yes, the XRD has shown there are peaks for both PEO and the op and cp MOF, this is expected as it's a physical mixture and it's the cumulative effect. The change in the op and cp form observed maybe due to the polymer blocking some pores of MOF.

3) Also, which MOF is used for the polymer-MOF composite? 1) methanol or activated-1?

Can you supply the pore size distribution and isotherm for porosimetry study on the pristine MOF and compared to the polymer-MOF composite, as excess PEO is used for the synthesis. So it will also point to whether the pores are occupied or not. The XRD of this pristine MOF should be included for all XRD figures for comparison to the polymer-MOF composite.

4) Fig. 3a should include all the XRD for the polymer-MOF composite, 1/PEO_{10k}, 1/PEO_{20k} and 1/PEO_{200k} to complement Fig. 3b. Bulk PEO xrd should be included as control.

5) The authors did an excellent job in studying the thermal degradation and crystallization process of the polymer-MOF composite (i.e. the 1/PEO_{2k1/1} and 1/PEO_{4M1/1}). I would like to dig deeper. The authors should further analyze the results using your already available data in SI Fig. S5 to S8.

i) represent the isoconversional Friedman plots for the 1/PEO_{2k1/1} and 1/PEO_{4M1/1} and their bulk PEO at different scan rate over the range $\alpha = 0.1$ to 0.7 (data from Fig. S7)

ii) regarding the relative crystallization, can represent the relative crystallization plots of the 1/PEO2k1/1 and 1/PEO4M1/1 under various cooling rates (data from Fig. S6) and compare to bulk

6) Any study on the melt crystallization kinetics and mechanism? To understand the crystallization process of 1/PEO2k1/1 and 1/PEO4M1/1, Ozawa, and Avrami or Tobin plots for the crystallization kinetics should be performed. The support for the schematic plot in Fig 5b is rather weak at the moment.

7) With the additional work in above Points 5 and 6, the author can further elaborate the thermal degradation and crystallization of the polymer-MOF composite, together with Fig. 5a and 5b. All these will point to the merits of using polymer-MOF composite, where this is lacking in the manuscript at the moment.

8) The terminology used "MOFaxane" in which the author used the analogy to polyrotaxane, but polyrotaxane has end-caps at the end of the polymers. However, there's no end caps for this polymer-MOF composite. Can the authors clearly define what MOFaxane mean? I'm not convinced to use this new terminology at the moment.

When I first read the word, MOFaxane, I had the first impression to relate to MXenes or MAXenes which are 2D structure; which clearly isn't the case and the structure is completely different.

The authors are speculating the new polymer-MOF composite would have similar characteristics as that of polyrotaxanes. But there's no solid evidence suggesting that the polymer-MOF composite has MOF sliding on the polymer chain, analogous to the cyclodextrin rings sliding on the axle chain for polyrotaxane (as on page 11 and 12 of the manuscript).

The author must think seriously in using the term MOFaxane, to prevent confusion and misrepresentation of the true property of this polymer-MOF composite.

9) The authors should update the list and make reference to other polymer threaded MOF publication: J. Am. Chem. Soc. 2014, 136, 20, 7209–7212 and ACS Energy Lett. 2021, 6, 11, 3769–3779

10) For mechanical property of the polymer-MOF composite, any results for 1/PEO2k1/1 and 1/PEO4M1/1? Should include for completion.

Point-to-Point Responses to Reviewer's Comments

We considered all comments from the reviewers and addressed their concerns in the revised manuscript. The responses to the reviewer's comments are summarized below. Considering both reviewer's concerns about the terminology of MOFaxane, we revised the related phrases with adding the definition of possible structure variations of MOFaxane (Figure 1) to avoid any confusion for readers. Additionally, minor grammatical errors were corrected, and English was polished up in the revised version. Please also find attached the revised manuscript (**Review-Only Material**) with all changes shown in **brown letters**. Now we believe that the revised manuscript is suitable for publication in *Nature Communications*.

For Reviewer 1:

The paper presents a simple approach to fabricate a so-called mechanically interlocked material, MOFaxane, in which the ultralong polymer chains thread microcrystals of metal-organic frameworks. Unlike most of the examples previously reported, the length of ultralong guest polymers here significantly exceeds the size of the MOF particles. It is worth noting that the cp-op structure change monitored with PXRD experiments proves the successful insertion of PEO chains into the nanopores of MOF and thus the authors thought the polythreaded structure is similar to the polyrotaxanes. Subsequently, the effects of the MOFaxane structure on the organic-inorganic hybrid materials' physical properties and bulk mechanical properties were investigated through a series of methods. Partial results are interesting and the project is worthy of investigation. However, there are several unreasonable explanations and more importantly, the proposed MOFaxane structure is not precise.

Some discussions may be helpful for the improvement of the manuscript before it is submitted again:

=>We thank this reviewer for his/her positive comments. We revised the manuscript according to this reviewer's thoughtful comments and concerns by adding new data and discussions to strengthen the evidence for the proposed structure. Please also find attached the revised version of Supporting Information that also encloses the important arguments.

1) The authors attempted to name the MOF composite by learning from the linear polyrotaxane. A rotaxane contains at least three elements: macrocycle, axle, stoppers. In the MOF composite, there are no stoppers. Therefore, strictly speaking, such analogy is not suitable. In fact, there are polypseudorotaxanes which are similar to the MOF composite in structure.

=>We seriously considered the comments from both Reviewer 1 and 2 about the terminology of our MOF-polymer composite. As suggested by the reviewers, the composite structure should be named after the existent rotaxane family by strictly following its classification rule. Considering the structural feature that has no stoppers, what we synthesized in the current work would be called "polypseudoMOFaxane" in the networking form. In the revised manuscript, we modified the definition statement of rotaxane family in the introductory part, and also clearly mentioned the polypseudorotaxane-like structure of the MOF composite obtained in this work. Further, to avoid

misunderstanding of readers, we defined the structure variations of MOFaxe family in the revised Figure 1. The MOFaxe family includes subclasses in analogy with rotaxane: pseudoMOFaxe, polyMOFaxe, and polypseudoMOFaxe (see Figure 1 given below). Although the present work only describes the synthesis of polypseudoMOFaxe, we believe that this becomes the first step for the realization of MOFaxe in the true form and also provide many prospects in the soft and responsive material designs owing to the numerous varieties of the MOFs as the functional rotors.

(Revised Figure)

Fig. 1 | Definitions and graphical representations of MOFaxe family. a, MOFaxe, b, pseudoMOFaxe, c, polyMOFaxe, and d, polypseudoMOFaxe. PseudoMOFaxe has no end cap on the polymer chains while MOFaxe has end caps by definition. PolyMOFaxe and polypseudoMOFaxe have network structures in which each polymer chain threads multiple MOF microcrystals, forming a topological network.

=>We revised the abstract and the introductory part of the main text as follows,

(Abstract)

“Mechanically interlocked architecture has inspired the fabrication of numerous molecular systems, such as rotaxanes, catenanes, molecular knots, and their polymeric analogues. However, to date, the studies in this field have only focused on the molecular-scale integrity and topology of its unique penetrating structure. Thus, the topological material design of such architectures has not been fully explored from the nano- to the macroscopic scale. Here, we propose a new supramolecular interlocked system, MOFaxe, comprised of long chain molecules penetrating a microcrystal of metal–organic framework (MOF). In this study, we describe the synthesis of polypseudoMOFaxe as the first example of the MOFaxe family. This has a polythreaded structure in which multiple polymer chains thread a single MOF microcrystal, forming a topological network in the bulk state. The novel topological crosslinking architecture is obtained by simply mixing polymers and MOFs, and displays characteristics distinct from those of conventional polyrotaxane materials, including suppression of unthreading reactions.”

(Introduction)

(Page 3, Line 1) “Rotaxane is a well-known family of interlocked molecules that consist of two major elements: a macrocyclic ring like a “rotor” and a threaded backbone string like an “axle” with bulky end caps^{1,2}.”

(Page 4, Line 1) “Here, we thread MOF microcrystals with ultralong polymer chains, thereby proposing a new class of interlocked system, “MOFaxane”, that utilizes MOFs as the rotors on the polymer chain axles (Fig. 1a). In analogy with rotaxanes, we can define subclasses of MOFaxane, which include pseudoMOFaxane and polymeric analogues (Fig. 1). In this study, we synthesized polypseudoMOFaxane as the first example of the MOFaxane family, which opens up new avenues to construct numerous interlocked materials using MOF crystals as the functional rotary elements.”

2) As for the polypseudorotaxanes made of cyclodextrins (CDs) and PEOs, the covering ratio can be calculated so that we know how many CDs threaded into the backbone of PEOs. Similarly, it is meaningful to estimate how many MOFs thread into the backbone of PEO4M. Moreover, how much is the driving force for the threading event and how is the influence of threading numbers of MOF on the threading efficacy. In other word, when the first MOF is threaded on the backbone of PEO, will it accelerate or slow down the second one for threading?

=>Unlike CD-based polypseudorotaxane, the accurate determination of covering ratio or the number of penetrations per PEO chain in the polypseudoMOFaxane is technically challenging because a single MOF microcrystal (**1**) has numerous pores in which PEO can evenly penetrate through. Nonetheless, we can estimate the provisional covering ratio based on the PEO loading amount and experimental gas adsorption data, using simple assumptions as follows. Firstly, we calculated how many pore entrances exist on the surface of MOF microcrystals. Based on the single crystal structure of **1** and the mean particle size (dimension = 640 nm × 640 nm × 80 nm), the number of pore entrances on the plate-like particle surface can be calculated to be 3.1×10^5 /particle. We assume that 100% covering ratio of PEO in **1** can be achieved at the point of the maximum loading capacity (0.23 g/g) of **1**. In other words, above 0.23 g/g PEO loading, all pores (i.e. pore entrances) of **1** are threaded by PEO chains. In this assumption, the covering ratio can be simply expressed as $0.23/x$ where x is the weight ratio of PEO to that of **1**. For example, the covering ratio of the **1**/PEO4M_{1/0.3} composite ($x = 0.30$) can be calculated to be $0.23/0.30 = 0.77$ (77%). By considering the number of pore entrances (3.5×10^5 /particle) and the average molar mass of each microcrystal (1.2×10^{10} g/mol), the number average penetration cycle per chain can be also calculated to be 380/chain.

In reality, however, it is unlikely that PEO chains penetrate all pores of **1** microcrystals due to the kinetic reason attributed to the ultralong length and entangled conformation. Indeed, we found that the **1**/PEO4M_{1/0.3} composite still has effective microporosity in the N₂ gas adsorption analysis that was newly performed in this revision cycle. The results were given in Supplementary Fig. 11. Considering the gas adsorption capacity of the pristine **1** in open form (205 mL at $P/P_0 = 0.9$), **1** microcrystals in the **1**/PEO4M_{1/0.3} composite preserves approximately 33% of the original porosity. In other words, 67% of the MOF pores (i.e. pore entrances) are involved in threading PEO4M in the composite. Based on this observation, the actual covering ratio of the **1**/PEO4M_{1/0.3} composite can be estimated to be 51%. By following this calculation method and the assumptions, we can also estimate the covering ratio of **1**/PEO4M_{1/1} composite to be 15%.

=>In the revised manuscript, we have included the following statement in the main text and a new

section in the Supplementary Information with the gas adsorption data.

(Main text)

(Page 10, line 1) “In analogy with the conventional cyclodextrin-based polyrotaxane, we estimated the percent coverage of the axial PEO with **1** microcrystals. Based on the actual pore occupancy evaluated by gas adsorption analysis, the averaged coverage is calculated to be 51% for **1/PEO4M**_{1/0.3} composite (See Supplementary Information and Supplementary Fig. 11).”

(Supplementary Information)

2. Estimation of the coverage of polypseudoMOFaxane

The percent coverage of PEO with **1** microcrystals was estimated based on the PEO loading amount and experimental gas adsorption analysis. Firstly, we calculated the number of pore entrances presenting on the crystal surfaces. Based on the single crystal structure of **1** and the mean particle dimension (640 nm × 640 nm × 80 nm), the number of pore entrances on the crystal surfaces was calculated to be 3.5×10^5 /particle. We assume that 100% coverage of PEO with **1** can be achieved at the point of the maximum loading capacity of PEO in **1** (0.23 g/g). In other words, all pores (i.e. pore entrances) of **1** are threaded by PEO chains at >0.23 g/g PEO loading. Under this assumption, the coverage can be simply expressed as $0.23/x$ where x is the weight ratio of PEO to that of **1**. For example, the coverage of the **1/PEO4M**_{1/0.3} composite ($x = 0.30$) can be $0.23/0.30 = 0.77$ (77%). In reality, however, it is unlikely that PEO chains penetrate all pores of **1** microcrystals due to the kinetic reason attributed to the ultralong length and entangled conformation. Indeed, the N₂ gas adsorption analysis showed that the **1/PEO4M**_{1/0.3} composite still has effective microporosity (Supplementary Fig. 11). **1** microcrystals in the **1/PEO4M**_{1/0.3} composite showed the adsorption capacity of ~68 mL (P/P₀ = 0.9) that is approximately 33% of that for the pristine **1** in open form (~205 mL). In other words, 67% of the MOF pores (i.e. pore entrances) are involved in threading **PEO4M** in the composite. Based on this result, the actual coverage of the **1/PEO4M**_{1/0.3} composite can be estimated as $(0.23/0.30) \times 0.67 = 0.51$ (51%). By following this estimation method and the assumptions, the coverage of **1/PEO4M**_{1/1} is also calculated to be 15%.

Supplementary Fig. 11.

a, N₂ adsorption isotherms of the pristine **1** (gray), **1/PEO4M**_{1/0.15} (green), and **1/PEO4M**_{1/0.3} (orange), measured at 77 K. **b**, Pore size distribution of the pristine **1** (gray), **1/PEO4M**_{1/0.15} (green), and **1/PEO4M**_{1/0.3} (orange), calculated by MP (micropore) method using the N₂ adsorption isotherms shown in the panel **a**. It was observed that the adsorption capacity decreases with increasing the PEO loading amount while the mean pore size is not significantly changed. This indicates that the decrease of adsorption capacity is ascribed to the decrease in the number of vacant pores by PEO threading.

=>Previously, we reported that the PEO insertion into MOF crystals is exothermic, thus enthalpy driven process (*Nat. Commun.* **2018**, *9*, 3635, *Chem. Sci.* **2021**, *12*, 12576–12586). Similar to the previous systems, we consider that this polypseudoMOF_{axane} formation event is also enthalpy driven process. To verify this thought, we newly performed a set of thermal analyses using differential scanning calorimetry (DSC) to observe the exotherm and measure actual enthalpy gain of the penetration event. To this end, we needed a reference substance that ideally has the same structure as **1** but does not show *cp-to-op* phase transition since this phase transition undergoes a large structure deformation that is endothermic process (*Angew. Chem. Int. Ed.* **2020**, *59*, 15325–15341). Thus, the observable heat flow in DSC measurements is a sum of these exo- and endothermic effects. In order to measure the endotherm upon the *cp-to-op* structure transition, we used nano-sized **1** crystals (~210 nm in diameter), hereafter termed **1'**, as the reference material that does not show the structure transition. Owing to the known shape-memory effect for the downsized MOF crystals (*Science* **2013**, *339*, 193–196), **1'** keeps open phase (**1'-op**) even after the removal of guest molecules at room temperature, which enabled us to determine the deformation energy (endotherm). By using this value, we could calculate the actual enthalpy of the threading event (thus, driving force), which is 32 kJ/mol per repeating unit of PEO. The enthalpy value is larger than that determined previously for another pillared-layer-type MOF system (7.7 kJ/mol) (*Nat. Commun.* **2018**, *9*, 3635), which can be due to smaller pore size of **1**. In the revised Supplementary Information, we created a new section to describe the experimental details and discuss about the driving force, as given below.

(Supplementary Information)

4. Discussion about the driving force of PEO penetration in **1**.

To investigate the thermodynamic background of PEO threading event, we measured a heat flow during the insertion process by DSC analysis. Since **1** shows structural change upon PEO insertion, we need to consider the *cp-to-op* deformation enthalpy (ΔH_{def}) of **1** in addition to the adsorption enthalpy of PEO inclusion (ΔH_{ads}). Hence, the observable DSC heat flow corresponds to $\Delta H_{\text{def}} + \Delta H_{\text{ads}}$. To have ΔH_{ads} value, which can be the main driving force of the threading event, we need to measure ΔH_{def} individually. To this end, we performed the following experiments using nano-sized **1** crystals (~210 nm in diameter), hereafter termed **1'**, as the reference material that shows the structure deformation without guest adsorption.¹

The 210 nm-size [Cu₂(bdc)₂(bpy)]_n (**1'**) was synthesized according to the literature procedure with slight modifications.¹ Due to the shape-memory effect by crystal downsizing, **1'** keeps open phase

even after removal of guest molecules at room temperature.¹ The metastable open phase (**1'**-*op*) spontaneously changes to the stable closed phase (**1'**-*cp*) when heated above 200 °C. This shape-memory effect is observed for **1** whose crystal size is below approximately 300 nm.¹ Using **1'** as the reference crystal, it is possible to estimate the enthalpy change of *cp*-to-*op* phase transformation without undergoing guest adsorption/desorption process. We synthesized **1'** containing CH₂Cl₂ as the guest solvent. Evacuating the **1'** nanocrystals at 100 °C resulted in the mixture of metastable (**1'**-*op*) and the stable (**1'**-*cp*) phases (**1'**-*cp/op*) (Supplementary Fig. 6). We note that the product becomes the mixture of close and open phases (**1'**-*cp/op*) since the sample of **1'** contains particles larger than 300 nm due to its original particle size distribution (Supplementary Fig. 4).

Supplementary Fig. 6.

a, **1'**-*cp* was immersed in CH₂Cl₂ and evacuated at 100 °C for 3 h, which gave the mixture of closed and open phases of **1'** (**1'**-*cp/op*). **b**, PXRD pattern of **1'**-*cp/op*. The peak intensity ratio of **1'**-*op* was 0.385. **c**, A relationship between PXRD peak intensity ratio of **1'**-*op* and mass fraction of **1'**-*op* for physical mixtures of **1'**-*op* and **1'**-*cp* with various mixing ratio.

To estimate phase transition enthalpy of **1'**, it is necessary to know the mass fraction of **1'**-*op* in the mixture (**1'**-*cp/op*). For this, we created a calibration curve using PXRD patterns of the mixtures which were prepared by purposely mixing pure **1'**-*op* and **1'**-*cp* with various weight ratios. The intensity ratio between **1'**-*op* peak at $2\theta = 8.2^\circ$ and **1'**-*cp* peak at 9.0° on the PXRD data was plotted as a function of the mass fraction of **1'**-*op*. For this analysis, we used the peaks at 8.2° and 9.0° as the indices of *op* and *cp* phases of **1'**, respectively, instead of the peaks at 16.4° and 16.9° (Figure 3). This is because the latter peaks showed severe overlapping due to the peak broadening caused by the small crystalline size. The calibration curve thus obtained showed a proportional trend

(Supplementary Fig. 6c). Using this calibration curve, the mass fraction of **1'**-*op* in the actual mixture (**1'**-*cp/op*) (Supplementary Fig. 6b) was calculated to be 0.44.

In the DSC heating curve of **1'**-*cp/op*, an exothermic peak was observed at 150–200 °C, which corresponds to the transition from **1'**-*op* to **1'**-*cp* (Supplementary Fig. 7a). This indicates that the *op*-to-*cp* backward transition is an endothermic process. By integrating the differential curve (Supplementary Fig. 7b) between the two DSC profiles of **1'**-*cp/op* and **1'**-*cp* in Supplementary Fig. 7a, the total heat released during the structure change from **1'**-*cp/op* to **1'**-*cp* was calculated to be 46 J/g. Therefore, considering the mass fraction of **1'**-*op* in **1'**-*cp/op* determined above, the actual enthalpy for the transition from **1'**-*cp* to **1'**-*op* (ΔH_{def}) was estimated to be 105 J/g.

Supplementary Fig. 7.

a, DSC heating curves of **1'**-*cp* (black) and **1'**-*cp/op* (blue). Scan rate: 1 °C/min. Exothermic peak in **1'**-*cp/op* corresponds to the phase transition from **1'**-*op* to **1'**-*cp*. **b**, Subtracted curve obtained by subtracting the black curve from the blue curve in the panel **a**. The dotted line corresponds to the baseline for the integration analysis.

The PEO insertion into **1** accompanies the endothermic structure deformation process (ΔH_{def}) and exothermic PEO adsorption process (ΔH_{ads}). Finally, we measured DSC heat flow of the PEO insertion into **1**, which shows *cp*-to-*op* structure change, using **PEO2k** as the guest. A mixture of **PEO2k** (4.02 mg) and **1** (2.88 mg) was placed in an Al pan and subjected to the DSC measurement (20 °C–100 °C, 1 °C/min). DSC peak was observed at 53 °C at which melting of **PEO2k** and the infiltration occurred simultaneously, releasing the heat (ΔH_{obs}) of 147 J per gram of PEO in total (Supplementary Fig. 8). As the bulk **PEO2k** melted at this temperature, the enthalpy of fusion (ΔH_{f}) should be also taken into account. ΔH_{f} was determined to be 193 J/g from the integration of the DSC heating curve of **PEO2k** alone (Supplementary Fig. 8). Therefore, ΔH_{ads} was calculated as $\Delta H_{\text{ads}} = \Delta H_{\text{obs}} - \Delta H_{\text{def}} - \Delta H_{\text{f}} = (147 \times 4.02) - (105 \times 2.88) - (193 \times 4.02) = -487$ mJ. Considering the maximum adsorption capacity of **1** (0.23 g/g), the actual amount of **PEO2k** adsorbed in **1** can be calculated as $2.88 \times 0.23 = 0.662$ mg. Therefore, ΔH_{ads} is calculated as $\Delta H_{\text{ads}} = -487 / 0.662 = -736$

J/g (per gram of adsorbed **PEO2k**), which is converted to -32 kJ/mol (per PEO repeating unit). As ΔH_{ads} is the negative value, the PEO threading is an exothermic, enthalpy driven process. The ΔH_{ads} value is larger than that observed previously for other MOF/PEO systems, e.g. -7.7 kJ/mol per repeating unit of PEO for the insertion into $[\text{Zn}_2(1,4\text{-ndc})_2(\text{ted})]_n$ (ndc = naphthalenedicarboxylate, ted = triethylenediamine).² The strong affinity of **1** and PEO can be attributed to the narrow pore size of **1**. It should be noted that the kinetic factor may also give a substantial effect on the overall penetration efficiency of the ultralong guests. Please also see the discussion in Section IV.2.

Supplementary Fig. 8.

DSC heating curves of **1/PEO2k** mixture (top) and **PEO2k** (bottom) in the heating rate of 1 °C/min. The broken lines correspond to the baselines for the integration. During the heating processes, the endothermic heats of 147 J and 193 J per gram of PEO were observed for **1/PEO2k** mixture and **PEO2k**, respectively.

=>We have added a following statement in the main text.

(Main text)

(Page 6, Line 7) “We investigated the thermodynamic origin of PEO insertion into **1** and calculated the adsorption enthalpy to be 32 kJ/mol (per repeating unit of PEO), which is larger than that measured for PEO adsorption into a typical pillared-layer-type MOF (See Supplementary Information and Supplementary Fig. 6–8).²¹”

=>We have cited a following literature in the main text (as Ref. 21) and the Supplementary Information (as Ref. 2).

Le Ouay, B. et al. Selective sorting of polymers with different terminal groups using metal-organic

frameworks. *Nat. Commun.* **9**, 3635 (2018).

⇒ We have added thermogravimetric analysis (TGA) data in the Supplementary Figure 3 as follows.

Supplementary Fig. 3

TG profile of DMF included (a) **1** (**1**⊃DMF) and **1'** (**1'**⊃DMF).

⇒ Although the influence of threading numbers of MOF on the threading efficacy is in our interest as well, it is challenging to clearly define the threading efficacy at this moment. We believe that the threading number can be dictated by the stabilization energy of the penetration event (i.e. the enthalpy determined above experiments) according to the thermodynamic law (Boltzmann distribution) when the system is at the equilibrium state. However, the threading efficacy can be largely affected by kinetic factors. We consider that the threading composite is still at the kinetically trapped state since the ultralong length and steric congestion of PEO chains increases energy barrier of the penetration process. Hence, to answer this reviewer's question, we need to consider more carefully the kinetic factors on the penetration efficacy; however, the ultralong nature of the present guest polymers hampers the experimental investigations. In this context, the second threading event may have lower energy barrier compared to the initial threading because of two possible reasons: (1) the polymer entanglement can be somewhat reduced just after coming out from the first MOF particle, (2) the polymer insertion into already-open pores is much easier than that is closed at the initial state. The detailed insertion and gate-opening process will be investigated in the future study. We are grateful to this reviewer for this insightful comment.

⇒ Because of the addition of above new discussions and experiments, we have added the descriptions about the synthesis and characterization of **1'** in Supplementary Information (Section I, Supplementary Fig. 1, 2, and 4).

3) The Figure 4c and Figure 5b are totally different on the description of the possible structure of the so called MOF_{axne}. What is the true structure? According to Figure 4c, it seems a linear structure, but for Figure 5b, it becomes a crosslinked network by using the MOF as a crosslinking point. From my point of view, I prefer to think the MOF_{axne} is a network and the numbers of threaded MOFs are not very high.

=>To make all representations consistent, we totally revised Figure 4 (given below). As described in the response to the comment 1, we newly added a definition of MOF_{axne} structures in the revised manuscript (Figure 1). Further, we clearly stated that the product of the present work is classified into the polypseudoMOF_{axne} with the networked structure to avoid misunderstanding of the readers. Please refer to the response to the comment 1.

(Main text)

Fig. 4 | Structural changes of **1 depending on the feed amount of PEO.** a, PXRD patterns for 1/PEO2k_{1/0.3}, 1/PEO2k_{1/0.15}, 1/PEO4M_{1/0.3}, and 1/PEO4M_{1/0.15}. b,c, Schematic of the infiltration of short PEO (b) and ultralong PEO (c) into MOF **1** using PEO amounts below and above the maximum loading capacity of **1**. When short PEO chains with an amount less than the maximum loading capacity is used (b, bottom), a limited number of particles of **1** (purple squares) convert into the *op* phase and quickly incorporate PEO (curved lines), whereas the remaining MOF particles that may have had less contact with the PEO chains remain in the *cp* phase. In contrast, ultralong PEO that is much longer than the particle size of **1** can penetrate and bridge across multiple crystals of **1** simultaneously. This polythreading structure leads to the complete opening of MOF **1** even though the amount of PEO used is less than the theoretical maximum (c, bottom).

=>According to the previous discussion about the covering ratio (response to the comment 2), we can estimate the number of threaded MOF pores per chain. However, this is not necessarily equal to the number of threaded MOF particles per chain since the penetration occurs in both intra- and inter-

particle basis. In this regard, we cannot discriminate intra- and inter-particle penetration numbers, unfortunately. Therefore, determination of the actual number of threaded MOF particles is technically difficult at present. To support the formation of poly-threaded complex (polypseudoMOFaxane), we performed particle size distribution measurements for the composite and provided the data in the revised Supplementary Information as Supplementary Fig. 12. The data is given below. For the **1/PEO4M** composite, the formation of complex which is larger than **1** and **PEO4M** alone were clearly observed. The agglomerated particles were also confirmed by AFM images on the dispersed composite deposited on the substrate (newly added Fig. 5 in the main text). These observations support the formation of poly-threaded structures. For AFM observations, please refer to the responses to the comment 9.

=>We have included the following descriptions regarding above discussions in the main text.

(Main text)

(Page 10, line 4) “As deduced from the coverage value, free PEO chains are likely present in between threaded microcrystals even for the high MOF loading composite. The bridging PEO chains tether multiple microcrystals together to form loose agglomerates in the solution phase. Indeed, the formation of such agglomerated structures were observed in the laser-scattering particle size distribution measurements (Supplementary Fig. 12) and AFM images of the dispersed samples (Fig. 5 and Supplementary Fig. 13 and 14).”

(Supplementary Information)

Supplementary Fig. 12.

Particle size distribution data for **1** (blue), **PEO4M** (gray), and **1/PEO4M_{1/1}** (orange) composite dispersed in CHCl_3 at room temperature. The samples were dispersed by stirring for 5 min and subjected to the laser-scattering particle size distribution measurements. The **1** particle alone and **PEO4M** solution showed the monodisperse peak at the size of $\sim 0.34 \mu\text{m}$ and $\sim 34 \mu\text{m}$ (peak top),

respectively. On the other hand, **1/PEO4M_{1/1}** composite showed the presence of much larger particles that are attributed to the formation of poly-threaded complex.

=>We have added AFM images of **1/PEO4M_{1/1}** composite in the main text as Fig. 5.

(Main text)

Fig. 5 | Morphologies of polypseudoMOFaxane. a,b AFM images of **1/PEO4M_{1/1}** composite deposited on a mica substrate. The **1/PEO4M_{1/1}** composite was dispersed by stirring for 5 min in chloroform (1 mg/mL) and deposited on a mica substrate by spin coating (2500 rpm, 5 sec) at 25 °C. **a**, 50 μm × 50 μm topographic image (scale bar, 10 μm). **b**, 20 μm × 20 μm topographic image (scale bar, 4 μm).

4) For the macroscopic mechanical characterization of the MOFaxane materials, the variations are too small to be utilized to explain the threading effect. As we know, solid additive can well reinforce the mechanical performance of polymers. Therefore, such minor changes in the parameters of mechanical properties can not be used directly. Moreover, the authors referred to the CDs-based sliding materials to explain their observation. If thing goes according to the author' hypothesis, the elastomeric modulus of the MOFaxane should decrease rather than increase a little bit. More suitable explanations and more measurements on the mechanical properties should be made for better understanding of the MOFaxane materials.

=>In order to better understand the mechanical properties, we conducted additional tensile tests on **1/PEO200k** composite films at higher temperatures (70 °C and 77 °C). PEO adopts a crystalline state below its melting temperature (63°C), and therefore the tensile properties measured at 25°C (Supplementary Fig. 24) may primarily reflect the mechanical features of PEO crystalline domains rather than the contribution of topological crosslinking. As pointed out by the reviewer, it is also possible that the filler effect is present. The tensile tests conducted at 70 °C and 77 °C, which are higher than melting temperature of PEO, clearly showed the differences between the pristine **PEO200k** and **1/PEO200k** composite films and provided a characteristic feature that can be explained by the polypseudoMOFaxane architecture. The typical stress-strain curves of the pristine **PEO200k** and **1/PEO200k** composite films at 70 °C and 77 °C are shown in Supplementary Fig. 25, and the elastic modulus are summarized in Supplementary Table 3.

Despite the measurement temperatures being above the melting temperature of PEO, the pristine **PEO200k** film exhibited an elastic modulus that can be attributed to the entanglement effect of long PEO chains. At 70°C, the **1/PEO200k** composite film displayed a 1.2-fold larger elastic modulus than that of the pristine **PEO200k** film (Supplementary Fig. 25a), which can be attributed to the contribution of topological crosslinking by MOFs. Interestingly, at 77 °C, this trend was reversed, with the elastic modulus of the **1/PEO200k** film becoming smaller than that of the pristine **PEO200k** film (Supplementary Fig. 25b). This finding rules out the possibility of the filler effect and suggests the presence of sliding motion that is promoted at such high temperature. Based on the slow *cp-to-op* deformation rate in Fig. 3b, we suppose that the sliding motion of **1** microcrystals on PEO is extremely slow and unlikely to be manifested in the mechanical tests at low temperature (~25 °C). On the other hand, at higher temperature (~77 °C), the sliding motion is promoted, causing the elastic modulus of the composite film to decrease. Additionally, the penetrating structure (pseudoMOFane structure) may reduce the relative degree of chain entanglement in the material because a portion of PEO chains is captured in the pores, which may also contribute to the material softening at higher temperatures.

To further verify our hypothesis, a PEO network crosslinked by covalent bonding needs to be compared with the polypseudoMOFane composite to confirm that the former has a higher elastic modulus due to the absence of a characteristic sliding relaxation mode. However, we are unable to synthesize such a control network for comparison due to technical limitations in precisely determining the number of topological crosslinks in the **1/PEO** composites.

=>We have revised the manuscript with reflecting the above argument. Please also see the related discussion in the comment 8 of Reviewer 2.

(Main text)

(Page 13, line 6) “Interestingly, the tensile tests conducted at a higher temperature of 77 °C revealed a reversed trend in the elastic modulus, where the composite film was less elastic compared to the pristine film (Supplementary Fig. 24 and Table 3). This temperature dependence suggests that the contribution of possible filler effect is ruled out, and it may reflect the sliding motion that gets promoted at this temperature. However, at this point, the sliding motion of MOF crosslinks on PEO axle was not clearly identified by other methods, including dynamic mechanical tests, due to technical limitations caused by the significantly slow diffusion of the PEO chain in the pore (See Supplementary Information).”

(Supplementary Information)

(Page 29, line 12) “Furthermore, PEO adopts a crystalline state below its melting temperature (63°C), and therefore the tensile properties measured at 25°C (Supplementary Fig. 24) may primarily reflect the mechanical features of PEO crystalline domains rather than the contribution of topological crosslinking. In this context, we conducted the tensile tests at 70 °C and 77 °C, which are higher than melting temperature of PEO (Supplementary Fig. 25 and Supplementary Table 3). The results clearly showed the differences between the pristine **PEO200k** and **1/PEO200k_{1/10}** composite films and

provided a characteristic feature that can be explained by the polypseudoMOFaxane architecture.

Despite the measurement temperatures being above the melting temperature of PEO, the pristine **PEO200k** film exhibited an elastic modulus that can be attributed to the entanglement effect of long PEO chains. At 70 °C, the **1/PEO200k_{1/10}** composite film displayed a 1.2-fold larger elastic modulus than that of the pristine **PEO200k** film (Supplementary Fig. 25a), which can be attributed to the contribution of topological crosslinking by MOFs. Interestingly, at 77 °C, this trend was reversed, with the elastic modulus of the **1/PEO200k_{1/10}** film becoming smaller than that of the pristine **PEO200k** film (Supplementary Fig. 25b). This finding rules out the possibility of the filler effect and suggests the presence of sliding motion that is promoted at such high temperature. From the slow *cp*-to-*op* deformation rate in Fig. 3b, it is considered the sliding motion of **1** microcrystals on PEO is extremely slow and unlikely to be manifested in such mechanical tests at low temperature (~25 °C). In our previous work, we reported that the effective diffusion constant of **PEO20k** in a MOF that has similar narrow channel with the diameter of 0.57 nm is $3.1 \times 10^{-14} \text{ m}^2/\text{s}$ (40 °C)⁵. This is 10^4 times slower than the diffusion constant of CD of the conventional polyrotaxane in solution, which has been experimentally determined to be $1.1 \times 10^{-10} \text{ m}^2/\text{s}$ using quasi-elastic neutron scattering (QENS) measurements (30 °C)⁶. At higher temperature (~77 °C), the sliding motion could be promoted, causing the elastic modulus of the composite film to decrease.”

=>In regard to the above revision, we have cited the following literatures in the revised Supplementary Information.

5.Oe, N., Hosono, N. & Uemura, T. Revisiting molecular adsorption: unconventional uptake of polymer chains from solution into sub-nanoporous media. *Chem. Sci.* **12**, 12576–12586 (2021).

6. Yasuda, Y. et al. Molecular Dynamics of Polyrotaxane in Solution Investigated by Quasi-Elastic Neutron Scattering and Molecular Dynamics Simulation: Sliding Motion of Rings on Polymer. *J. Am. Chem. Soc.* **141**, 9655–9663 (2019).

=>We have added following figure and table in the Supplementary Information.

(Supplementary Information)

Supplementary Fig. 25.

Typical stress-strain curves of the **PEO200k** (black) and **1/PEO200k_{1/10}** (orange) films at **a**, 70 °C, and **b**, 77 °C.

Supplementary Table 3.

Temperature dependence of elastic modulus of **PEO200k** and **1/PEO200k_{1/10}** film ($N = 3$).

Temperature	PEO200k	1/PEO200k_{1/10}
25 °C	319 ± 58 MPa	492 ± 47 MPa
70 °C	113 ± 15 MPa	139 ± 33 MPa
77 °C	48.4 ± 17.0 MPa	13.3 ± 13.2 MPa

5) Fig. 2b shows the gate-opening behavior from 1-*cp* to 1-*op* phase corresponding to the peaks of PXRD at 16.9° and 16.4°, respectively. Could you please plot the corresponding crystal forms [1-10] and [200] at Fig. 2b to make it a little bit more intuitive?

⇒ We calculated the fraction of 1-*op* phase based on the PXRD data to monitor the PEO infiltration process. Due to the overlapped diffractions of 1-*op* and 1-*cp* phases at the intermediate state, we carried out the peak deconvolution analysis on the observed PXRD pattern. The reason why we selected 16.9° and 16.4° peaks for this analysis is because they are well separated, thus making the peak deconvolution much reliable compared to when using other diffraction peaks. Therefore, the selection of the peaks at 16.9° and 16.4°, which correspond to (1-10) of 1-*op* and (200) of 1-*cp*, respectively, is merely due to the technical reason, and we believe that other peaks can also be used to calculate the 1-*op* fraction if the peak deconvolution analysis works reliably.

6) At the bottom of page 5, ‘heated at 120 °C and annealed for 90 min under nitrogen atmosphere’ is not mentioned in the Methods. Please check it and fill in the missing pieces.

⇒ We have included a missing subsection in the Methods and provided details of the direct (molten phase) PEO insertion procedure as follows. The insertion process was performed using the XRD-DSC apparatus.

(Methods)

“Procedure for the direct insertion of molten PEO into 1. Direct insertion of molten PEO into **1** was performed on an XRD-DSC module attached to a Rigaku model Smart Lab X-ray diffractometer. A mixture of **1** and powdery PEO (1/PEO = 1/2, wt/wt, 4 mg in total) was placed on the aluminum sample plate (Rigaku). The temperature was controlled by using the XRD-DSC module with the following ramp and hold program: 25 °C | 10 °C/min (ramp) → 120 °C (target temperature) | 90 min (hold). To prevent undesirable oxidation and decomposition of PEO, 2,6-di-*tert*-butyl-4-methylphenol (3 wt%) was added to PEO and PXRD measurements were performed under nitrogen flow (120 mL/min). While heating to the target temperature, PEO melted. Upon heating, the PXRD patterns were recorded at a regular time interval. The time evolution plots of the fraction of 1-*op* displayed in Fig. 3b were generated based on the height ratio of the two representative diffraction peaks of the 1-

cp and **1-op** phases, (1-10) and (200) at $2\theta = 16.9^\circ$ and 16.4° , respectively, for each time-resolved PXRD pattern.”

7) The successful insertion of PEO4M was achieved in the presence of a sacrificial guest. For further verifying PEO4M's infiltration and exploring the internal structure, could you please evaluate the pore size distribution of the closed-MOF, MOF-CHCl₃ and MOF-PEO4M with nitrogen sorption experiment?

=>In the revised manuscript, we have newly provided N₂ gas adsorption isotherms and pore size distribution data for **1**, **1/PEO4M_{1/0.15}**, and **1/PEO4M_{1/0.3}** composites in Supplementary Fig. 11. Please refer to our responses to the comment 2. Upon gas adsorption, **1-cp** changes to **1-op**, thus showing significant gas adsorption and microporosity.²² The composites, **1/PEO4M_{1/0.15}**, and **1/PEO4M_{1/0.3}**, showed 38% and 67% decrease in the adsorption capacity from that of the pristine **1** due to the PEO threading. As stated in the responses to the comment 2, the **1/PEO4M_{1/0.3}** composite, which has enough PEO loading to fill all pores of **1**, still has microporosity (20% of the pore remains unthreaded). We consider that the composite with ultralong PEO has the kinetically trapped state due to large diffusional barrier. This feature is well reflected in the pore size distribution data obtained by MP method on the respective isotherms (Supplementary Fig. 11b). It was observed that the adsorption capacity decreases depending on the loading amount of PEO while the mean pore size is not significantly changed for all samples. This indicates that the decrease of adsorption capacity is ascribed to the decrease in the number of vacant pores by PEO threading. For the gas adsorption measurements, the samples need to be pre-activated in vacuum. Thus, the porosity analysis of the solvent loaded sample (MOF-CHCl₃) cannot be performed.

8) For 1/PEO2k, the co-existence of 1-op and 1-cp phase was observed by regulating the proportion of MOF and polymer chains. In order to gain insight into structure and minimize internal tangles, it's necessary to screen the ratio to find the critical point of PEO4M's content, below which the co-existence of 1-op and 1-cp phase in 1/PEO4M will occur.

=>We performed additional experiments for **1/PEO4M_{1/0.10}** and **1/PEO4M_{1/0.05}** to find out the lowest critical point of **PEO4M** content below which **1-cp** appears. The results were summarized in newly added Supplementary Fig. 10. We found that the 1/0.10 (wt/wt) composite, **1/PEO4M_{1/0.10}**, shows coexistence of **1-cp** phase, indeed. Therefore, approx. 1/0.10 ratio is the critical point at which all MOF microcrystals encounter PEO chain to be penetrated. At 1/0.05 loading (**1/PEO4M_{1/0.05}**), the diffraction peak intensity of **1-op** decreased. We calculated the fraction of **1-op** phase of the **1/PEO4M_{1/0.05}** composite using the calibration curve given in Supplementary Fig. 6, and found that the **1-op** fraction is 81.5% even at 1/0.05 **PEO4M** loading. This high gate-opening efficacy of **PEO4M** clearly underpins the formation of the polythreading configuration.

=>We have included the following statement and figure regarding above discussions in the main text and Supplementary Information, respectively.

(Main text)

(Page 9, line 1) “Interestingly, we found that even 1/0.05 (**1/PEO4M_{1/0.05}**) showed a high fraction of **1-*op*** phase (81.5%, Supplementary Fig. 10).”

(Supplementary Information)

Supplementary Fig. 10.

PXRD profiles for the **1/PEO4M_{1/0.30}** (orange), **1/PEO4M_{1/0.15}** (green), **1/PEO4M_{1/0.10}** (blue) and **1/PEO4M_{1/0.05}** (purple). **1-*cp*** phase appeared in **1/PEO4M_{1/0.10}**, indicating approx. 1/1.10 ratio (wt/wt) is the critical point at which all MOF microcrystals encounter PEO chain to be penetrated with **PEO4M**. At 1/0.05 loading (**1/PEO4M_{1/0.05}**), the diffraction peak intensity corresponding to the **1-*op*** phase decreased. The fraction of **1-*op*** phase in the **1/PEO4M_{1/0.05}** composite was determined using the calibration curve given in Supplementary Fig. 6. The **1-*op*** fraction was 81.5% even at 1/0.05 **PEO4M** loading. This high gate-opening efficacy of **PEO4M** clearly underpins the formation of the polythreading configuration.

9) PXRD alone is not a powerful method to prove the rotaxane-like conformation in the MOF_{axane} demonstrated in Fig. 1. It is also possible that the conformation of MOF_{axane} is a tightly packed spherical structure due to strong chain entanglement effect and the interactions between MOF particles. Please use some more intuitive characterization methods to prove the rotaxane-like conformation.

⇒We agree with this reviewer. The polypseudoMOF_{axane} network could have aggregated form in the dispersed state in solution. To confirm this, we performed particle size distribution measurements and provided the data in Supplementary Fig. 12. Please also refer to our responses to the comment 3. Indeed, the polypseudoMOF_{axane} (**1/PEO4M_{1/1}**) showed extremely large particle sizes that were observed in the dispersed solution of **1** or **PEO4M** alone. This result supports the topologically crosslinked network formation. Further, to investigate the morphological feature of this network, we

performed AFM imaging analysis for the **1/PEO4M_{1/1}** composite. Results and discussions about the AFM analyses were given in the response to the comment 2 of Reviewer 2 as this topic is also related to the Reviewer 2's question about the direct observation of the composite.

10) Compared with PEO2k, what're the reasons for the obvious decrease of 1/PEO2k's effective activation energy after 50% crystallinity shown in Fig. 5a?

=>At present, the reason for the decrease in activation energy of **1/PEO2k** in the high crystallinity region is not clearly understood. As one of possible reasons, we could consider a certain template effect of the MOF microcrystals. The MOF pores have a crystalline ordered structure. This may assist the organization of PEO chains to proceed crystallization even in the high crystallinity region. Although this finding is very interesting and worth to be investigated in detail, we consider that the phenomenon might deviate from the main scope of this paper. Therefore, it will be studied and reported in the future. We are grateful to this Reviewer for this insightful comment.

11) Please discuss the internal mechanism by which the effective activation energy of 1/PEO4M decreases first and then increases.

=>The crystallization behavior was investigated for the **1/PEO4M_{1/1}** composite that contains excess amount of PEO (1/1, wt/wt). We consider that the nucleation and crystallization occur in the excess PEO domains (outside of MOF crystals) in the initial stage of crystallization (low crystallinity region, $\alpha < 30\%$). Thus, this stage of crystallization is not affected by the presence of MOF crystals, resulting in the same trend of the activation energy for all samples in this crystallinity region ($0\% < \alpha < 30\%$). The trend that the activation energy decreases as the crystallinity increases suggests that the crystallization process is accelerated by the progressive formation of its own nucleation centers, which is observed in common polymer crystallization process. The short PEO/MOF composite, **1/PEO2k_{1/1}**, has the same situation because the crystallization process mainly occurs in the sufficiently mobile chains existing outside of MOF crystals. In this composite, **PEO2k** chains are encapsulated entirely and sequestered in the MOF crystals, thus not involved in the crystallization process. However, the behavior is reversed above $\alpha < 50\%$ for the ultralong PEO/MOF composite due to the polypseudoMOFaxane structure in which multiple MOF microcrystals are tethered by bridging PEO chains. This configuration hampers the reorganization of chains in the high crystallinity region, thereby inducing pronounced retardation (i.e. high activation energy) of the crystallization process.

=>We have revised corresponding text with adding some explanations in the main text.

(Main text)

(Page 11, line 5) "In the initial stage of crystallization, the activation energy decreases as the crystallization proceeds for all PEOs and composites. This suggests that the crystallization process is promoted by the progressive formation of its own nucleation centers in the PEO domain. However, only **1/PEO4M_{1/1}** demonstrates a significant increase in its activation energy as crystallization

progressed, which is the opposite trend to the other samples, the pristine PEOs (**PEO2k** and **PEO4M**) and **1/PEO2k_{1/1}**.”

12) In the unthreading experiment, we don't know if it was chain entanglement or multiple interactions between **1** and **PEO4M** that prevented **1/PEO4M**'s unthreading reaction. Let's come back to question (4), the unthreading experiment should be taken under the circumstances that the internal tangles are minimized.

=>We thank this reviewer for the helpful comment. In the original manuscript, we described the results of washing experiment on **1/PEO4M_{1/1}**, which contains an excess amount (1/1, wt/wt) of **PEO4M**. To determine which factor (chain entanglement or physical interaction) dominantly prevents the unthreading reaction, we conducted an additional washing experiment on **1/PEO4M_{1/0.3}** (1/0.3, wt/wt) for comparison. The internal chain entanglements in **1/PEO4M_{1/0.3}** are expected to be lower than those in **1/PEO4M_{1/1}**, as deduced from the estimated PEO coverage values: 51% for **1/PEO4M_{1/0.3}** and 15% for **1/PEO4M_{1/1}**. Please refer to our response to comment 2 for more information. We determined the amount of PEO remaining in the washed composite by analyzing the sample with ¹H NMR after digesting it in D₂O/EDTA-4Na solution (see Methods). The results showed that **1/PEO4M_{1/0.3}** retained only 0.06 g/g of PEO after washing with DCM for 6 h (see Supplementary Fig. 21). This amount is significantly lower than the amount of PEO retained in **1/PEO4M_{1/1}** (0.25 g/g) after the washing treatment. This result suggests that chain entanglements play a significant role in preventing the unthreading reaction. Therefore, we suppose that the present polypseudoMOF_{axane} structure is mainly stabilized by kinetic factors that rely on the entanglements of ultra-long PEO chains penetrating the MOF particles.

=>We have revised the main text as follows. The new data and discussions have been added in the Supplementary Information.

(Main text)

(Page 12, Line 1) “Interestingly, the washing experiment on **1/PEO4M_{1/0.3}**, in which the inter/intra-molecular entanglements of PEO chains are expected to be lower than that of **1/PEO4M_{1/1}** due to the higher coverage value, also showed a large loss of **PEO4M** (see Supplementary Information and Supplementary Fig. 21). These results suggest that chain entanglements play a significant role in preventing the unthreading reaction. Therefore, it can be considered that the present polypseudoMOF_{axane} structure is mainly stabilized by kinetic factors that rely on the entanglements of ultra-long PEO chains penetrating the MOF particles.”

(Supplementary Information)

(Page 25, line 9) “In order to better understand the PEO-holding mechanism, we measured the amounts of PEO remaining in **1** after the 6 h-washing treatment for **1/PEO2k_{1/1}** and **1/PEO4M_{1/1}**. In addition, to determine which factor (chain entanglement or physical interaction) prevents the unthreading reaction, **1/PEO4M_{1/0.3}** was also examined for comparison to **1/PEO4M_{1/1}**. The entanglements of PEO chains in **1/PEO4M_{1/0.3}** are expected to be lower than those in **1/PEO4M_{1/1}**,

as deduced from the estimated PEO coverage values: 51% for **1/PEO4M_{1/0.3}** and 15% for **1/PEO4M_{1/1}** (see Section IV.2). The result showed that **1/PEO4M_{1/0.3}** retained only ~0.06 g/g of PEO after washing (Supplementary Fig. 21) whereas **1/PEO4M_{1/1}** retained 0.25 g/g after the same treatment. This suggests that preventing the unthreading reaction is mainly due to the chain entanglements, rather than the physical interactions between **1** and PEO.”

(Supplementary Information)

Supplementary Fig. 21.

The amount of PEO in the composites **1/PEO2k_{1/1}**, **1/PEO4M_{1/0.3}**, and **1/PEO4M_{1/1}** after the DCM-washing for 6 h. The amount of PEO was determined by ¹H NMR analysis on the digested sample.

- 13) The dynamic motion of MOF particles sliding on the axle chain may bring the material toughness enhancement. Please calculate and compare the toughness of several films according to their typical stress–strain curves shown in Supplementary Fig. 11 to explore the possible sliding motion.

=>Typical mechanical performances (elastic modulus, stress at yield, stress at break, and elongation at break) of the **1/PEO** composite films at 25 °C were provided in Supplementary Table 2. To explore the possible sliding motion, we conducted tensile test at higher temperatures of 70 °C and 77 °C. Please see the additional results and discussions described in the response to the comment 4. Although identifying the sliding motion is technically challenging by mechanical property analysis because of extremely slow diffusion of PEO chain in the MOF pores, we believe that further optimization of MOF structures will allow us to detect the sliding motion in practical time scale and reasonable temperature conditions. We have started synthesis of the polypseudoMOF_{axane} using non-crystalline polymers with low T_g (glass transition temperature), in which the polymer diffusion in the

pore is expected much faster at room temperature region, facilitating further exploration of the characteristic mechanical properties arising from the sliding relaxation mode. We will report the studies focusing on the mechanical properties in our subsequent paper in comprehensive way.

The Supplementary Table 2 provides information on the typical mechanical properties of the **1**/PEO composite films, including the elastic modulus, stress at yield, stress at break, and elongation at break at 25°C. To investigate the possibility of sliding motion, we performed a tensile test at higher temperatures of 70°C and 77°C. For further details on these results, please refer to our response to comment 4 of Reviewer 1 and comment 8 of Reviewer 2. Although detecting the sliding motion through mechanical property analysis is technically difficult due to the slow diffusion of PEO chains in the MOF pores, we are confident that by optimizing the MOF structures, we will be able to observe this motion under practical conditions and at reasonable temperature ranges. In order to explore the unique mechanical properties arising from the sliding relaxation mode, we have started synthesizing polypseudoMOFaxane using non-crystalline polymers with low T_g (glass transition temperature), which are expected to diffuse more quickly in the pore at room temperature. We will comprehensively report our subsequent studies on the mechanical properties of polypseudoMOFaxane in a subsequent paper.

For Reviewer 2:

The authors reports threading the PEO into the MOF crystal and performed detailed study on the thermal degradation and the rate of crystallization of the polymer-MOF composite (1/PEO_x). This work is significant as the field is lacking in the formation mechanism in the polymer threaded MOF.

=>We thank this reviewer for his/her encouraging comments. Please find our responses to the reviewer's concerns below.

1) In terms of the structure and morphology of the polymer-MOF composite, SEM and or TEM images of the pristine MOF, and after loading PEO (1/PEO2k_{1/1} and 1/PEO4M_{1/1}), as the major results are for these two composites. Thus, more information on the structure and morphology on the composites.

=>Due to the limit of imaging resolution and material damages by electron dosing, our attempts on SEM and TEM observations of the polymer-MOF composites were failed to give meaningful information for their morphology. Alternatively, we performed AFM imaging analysis on the 1/PEO2k_{1/1} and 1/PEO4M_{1/1} composites. The 1/PEO4M_{1/1} composite was dispersed by stirring 5 min (25 °C) in chloroform (1 mg/mL) and deposited on a mica substrate by spin coating (2500 rpm, 5 sec). The deposited particles were observed by AFM in non-contact tapping mode.

Topographic AFM images of the deposited 1/PEO4M_{1/1} are given in the newly added Fig. 5 in the main text (Please refer to our response to the comment 3 from Reviewer 1). In comparison to the pristine **1** microcrystals (an AFM image was given in Figure 2a of the main text), 1/PEO4M_{1/1} showed obvious agglomerations of the crystals. Interestingly, in the 1/PEO4M_{1/1} composite, the microcrystals of **1** form loosely tethered each other to form gatherings, rather than forming massive aggregates. This agglomeration formation of 1/PEO4M_{1/1} is consistent with the results of the particle-size distribution measurements (Supplementary Fig. 12). This morphological feature is in good agreement with what we envisioned for the polypseudoMOF_{axane} (Figure 1d) in which polymer chains are weaving and tethering multiple MOF particles, forming the loose network structures. The loose networking morphology is also reasonable when considering the PEO covering ratio of 1/PEO4M_{1/1} composite, which is calculated to be 18% according to the discussion in the comment 2 of Reviewer 1. It should be noted that PEO4M chains were observed in the background of the AFM images since the amount of PEO is in an excess to the MOF capacity in the 1:1 composite.

Interestingly, AFM for 1/PEO4M_{1/0.3} composite (1:0.3) visualized more intuitive morphology reflecting the polypseudoMOF_{axane} structure. The AFM images of the dispersed 1/PEO4M_{1/0.3} were given in Supplementary Fig. 13. Radial dendritic PEO crystals were observed at the periphery of each agglomerated particle of the composite. The intriguing morphology that all PEO radial domains were localized in contact with the MOF particles and neither individual PEO crystals nor MOF particles were observed. This morphological feature supports the polypseudoMOF_{axane} structure in which extremely long PEO chains topologically bind multiple **1** microcrystals. For comparison, we performed AFM imaging for an instant mixture of **1** and PEO4M. **1** was dispersed in chloroform (0.5 mg/mL) by sonication for 20 min prior to mixing. The **1** suspension and PEO4M solution (0.5 mg/mL

in chloroform) were mixed by stirring for 5 min (25 °C) to have the instant mixture in 1/1, wt/wt, ratio. The mixture was deposited on a mica substrate by spin coating (2500 rpm, 5 sec). The AFM images of the mixture showed a different morphology (Supplementary Fig. 14). No localization of PEO chains was observed. The PEO chains appeared as the homogeneous background with uniformly distributed dot-like nanocrystals. These observations support the penetration of PEO chain through **1** crystals in the composites. Although this reviewer suggested testing **1/PEO2k_{1/1}** composite, we did not perform AFM imaging of the composite since dispersing the composite in chloroform causes unthreading of **PEO2k** from **1** (Please refer to the response to the comment 12 of Reviewer 1 and Supplementary Fig. 21). This will end up with the mixture of **1** and **PEO2k** and lead to the same result with the above experiment.

=>We have added a new section about the AFM analysis in the Supplementary Information as follows.

(Supplementary Information)

4. AFM imaging of polypseudoMOF_{axane}

The AFM imaging for **1**, **1/PEO4M_{1/0.3}**, and **1/PEO4M_{1/1}** were performed as follows. Each sample was dispersed by stirring for 5 min (25 °C) in DMF or chloroform (1 mg/mL) and deposited on a mica substrate by spin coating (2500 rpm, 5 sec). The deposited particles were imaged using Asylum Research model MFP-3D Origin operated in non-contact tapping mode. A silicon cantilever (OMCL-AC240TS, Olympus) with a spring constant ranging from 0.6 to 3.5 N/m (resonant frequency of 50-90 kHz) was used and calibrated by the thermal fluctuation method. Igor Pro software (WaveMetrics) was used for all of the data acquisition and analysis.

While the AFM image of the pristine **1** showed individually dispersed microcrystals (Fig. 2a), the **1/PEO4M_{1/1}** composite showed obvious agglomerations of the crystals (Fig. 5). In the **1/PEO4M_{1/1}** composite, the microcrystals of **1** form loosely tethered each other to form gatherings, rather than forming massive aggregates. This agglomeration formation of **1/PEO4M_{1/1}** is consistent with the results of the particle-size distribution measurements (Supplementary Fig. 12). This morphological feature is in good agreement with what we envisioned for the polypseudoMOF_{axane} (Figure 1d) in which polymer chains are weaving and tethering multiple MOF particles, forming the loose network structure. This morphology is also reasonable when considering the PEO coverage of the **1/PEO4M_{1/1}** composite, which is estimated to be 15% (Section IV.2). It should be noted that the most of **PEO4M** chains were observed in the background as a thin film homogeneously covering the substrate since the amount of **PEO4M** is in an excess to the MOF capacity in this 1/1 composite.

Interestingly, the 1/0.3 composite, **1/PEO4M_{1/0.3}**, showed more intuitive morphology supporting the polypseudoMOF_{axane} structure (Supplementary Fig. 13). For the **1/PEO4M_{1/0.3}** composite, dendritic PEO crystals were observed at the periphery of each particle. This intriguing morphology that all PEO radial domains are localized in contact with **1** microcrystals, and neither individual PEO crystals nor MOF particles were observed. This morphological feature supports the polypseudoMOF_{axane} structure in which extremely long PEO chains topologically bind multiple **1** microcrystals. For comparison, we performed AFM imaging for an instant mixture of **1** and **PEO4M**. **1** was dispersed

in chloroform (1 mg/mL) by sonication for 20 min prior to mixing. The **1** suspension and **PEO4M** solution (1 mg/mL in chloroform) were mixed by stirring for 5 min (25 °C) to have the instant mixture in 1/1, wt/wt, ratio. The mixture was deposited on a mica substrate by spin coating (2500 rpm, 5 sec). The AFM image of the mixture showed slight agglomerations of **1** microcrystals, but with different morphology (Supplementary Fig. 14). No localization of PEO chains was observed. The PEO chains appeared as the homogeneous background with uniformly distributed dot-like nanocrystals. These observations suggest that the polypseudoMOF_{axane} structure is not effectively formed just by instant mixing in solution phase. To obtain the polythreading structure, successive evaporation of the sacrificial guest solvent (chloroform) at high temperature is indispensable process due to the slow diffusion of PEO in **1**.

Supplementary Fig. 13.

Topographic AFM images of **1/PEO4M_{1/0.3}** composite deposited on a mica substrate. **a,b,c**, The AFM images highlighting separated agglomerates, and **d,e,f**, those in different height contrast, respectively.

The contrast of **d,e,f** is adjusted to visualize the radial PEO crystals surrounding each agglomerate.

Supplementary Fig. 14.

Topographic AFM images of the **1** and **PEO4M** mixture (1/1, wt/wt) deposited on a mica substrate. **a**, The AFM image highlighting separated agglomerates, and **b**, that in different height contrast. The contrast of **b** is adjusted to visualize the location of PEO domains. PEO chains were observed in the background as uniformly distributed dot-like nanocrystals.

- 2) From the synthesis, PEO is inserted in MOF by molten phase insertion. Could the authors explain how the polymer thread into the MOF pore? Detail discussion should be included in the maintext, in addition to Fig. 5b.

Also additional characterization like cross-section TEM maybe needed to prove PEO is threaded into MOF.

At the moment, it seems to be either the MOF are filling the interspace between the PEO polymer chain, or the bulk PEO polymer is wrapping the MOF, or coating the MOF. Yes, the XRD has shown there are peaks for both PEO and the op and cp MOF, this is expected as it's a physical mixture and it's the cumulative effect. The change in the op and cp form observed maybe due to the polymer blocking some pores of MOF.

=>The general mechanism of molten-phase PEO infiltration into MOFs has been investigated and discussed in our previous reports (*J. Phys. Chem. C* **2015**, *119*, 21504–21514, *Nat. Commun.* **2018**, *9*, 3635, *J. Am. Chem. Soc.* **2020**, *142*, 3701–3705). In principle, the polymer infiltration into MOF pores undergoes tremendous entropic penalty. However, if there is sufficient enthalpy gain to compensate the entropy loss, the infiltration process proceeds. In this case, the infiltration is enthalpy driven and exothermic process (Please also refer to our responses to the comment 2 from Reviewer 1). The insertion of the polymer chain occurs from the extremity of the chain. In general, the diffusion process of the polymer chain in the MOF pores is very slow, limiting the overall rate of the infiltration event. To clearly explain the MOFaxane structure we have obtained in this study, we revised Figure 1 to define the structure variations of MOFaxane family in the revised manuscript. In this study, we synthesize polypseudoMOFaxane as the first example of MOFaxane family.

=>Despite our attempts of TEM, the PEO chain penetrating **1** crystals was not visualized. Since **1** is very sensitive to the electron beam, the crystal structure of **1** was lost immediately during the imaging. This feature makes the atomic-resolution observation difficult. Further, since PEO consists of only light elements, H, C and O, it is technically difficult to have sufficient image contrast that allows to identify single chain penetrating the sub-nanometer pores. Alternative to the TEM observation, we performed AFM imaging of the composites, and images are given in Figure 5 and Supplementary Fig. 13. Please refer to our responses to the comment 1. We believe that the AFM results well support the formation of polypseudoMOFaxane.

=>We do not think that *cp-to-op* structure change of **1** is induced by the polymers blocking some pores of **1** because the *cp-to-op* transition rate depends on the molecular weight of PEO (Figure 3). Further, PEO chains with large end-capping groups did not induce the *cp-to-op* structure change (Supplementary Fig. 9), which clearly indicate that the insertion of PEO chain into the pore induces the structure change, thus initiates the infiltration event.

3) Also, which MOF is used for the polymer-MOF composite? 1⊃methanol or activated-1?

Can you supply the pore size distribution and isotherm for porosimetry study on the pristine MOF and compared to the polymer-MOF composite, as excess PEO is used for the synthesis. So it will also point to whether the pores are occupied or not. The XRD of this pristine MOF should be included for all XRD figures for comparison to the polymer-MOF composite.

=>We used activated-**1** (namely, **1-cp**) for the preparation of polymer/MOF composites. Regarding the pore size distribution and isotherm, we performed N₂ gas adsorption measurements and have provided the data in the revised Supplementary Information as Supplementary Fig. 11. Please also refer to our answer to the comment 2 from Reviewer 1. We found that **1/PEO4M**_{1/0.3}, which has enough PEO to fill all the pores (as maximum capacity of **1** is ~0.23 g/g), still shows microporosity (Supplementary Fig. 11). This indicates that the composite has unoccupied pores since the threading efficacy is not perfect due to the kinetic reason. This observation provided a useful aspect of the covering ratio that is calculated to be 51% for **1/PEO4M**_{1/0.3}.

=>As per this reviewer's suggestion, we have added the PXRD data of pristine **1** (**1-cp**) and **1-op** phases instead of their simulated patterns in Figure 4. Figure 4 was also revised in response to the comment from Reviewer 1. Please refer to our answer to the comment 8 from Reviewer 1.

4) Fig. 3a should include all the XRD for the polymer-MOF composite, **1/PEO10k**, **1/PEO20k** and **1/PEO200k** to complement Fig. 3b. Bulk PEO xrd should be included as control.

=>We have added the PXRD data of **1/PEO10k**, **1/PEO20k**, and **1/PEO200k** to Figure 3a (given below). We did not include the PXRD data of bulk PEO because it melts and gives no diffraction at the monitoring temperature of 120 °C.

(Main text)

Fig. 3 | Structural changes in 1 induced by PEO infiltration. **a**, Powder X-ray diffraction (PXRD) profiles for the mixture of 1/PEO2k (blue), 1/PEO10k (green), 1/PEO20k (yellow), 1/PEO200k (purple), and 1/PEO4M (orange) (mixing ratio, 1/PEO = 1/2, wt/wt) after the heating at 120 °C for 90 minutes under nitrogen atmosphere. Black and gray lines denote the PXRD patterns of 1 including methanol as the guest molecule, showing 1-op phase, and 1 in the evacuated state, showing 1-cp phase, respectively. **b**, Time evolution plots of 1-op fraction of in contact with molten PEO with MW of 2,000 (PEO2k) (blue), 10,000 (PEO10k) (green), 20,000 (PEO20k) (yellow), 200,000 (PEO200k) (purple) and 4,000,000 (PEO4M) (orange), monitored in situ at 120 °C under nitrogen atmosphere.

- 5) The authors did an excellent job in studying the thermal degradation and crystallization process of the polymer-MOF composite (i.e. the 1/PEO2k1/1 and 1/PEO4M1/1). I would like to dig deeper. The authors should further analyze the results using your already available data in SI Fig. S5 to S8.
- i) represent the isoconversional Friedman plots for the 1/PEO2k1/1 and 1/PEO4M1/1 and their bulk PEO at different scan rate over the range $\alpha = 0.1$ to 0.7 (data from Fig. S7)
 - ii) regarding the relative crystallization, can represent the relative crystallization plots of the 1/PEO2k1/1 and 1/PEO4M1/1 under various cooling rates (data from Fig. S6) and compare to bulk

=>We appreciate this useful suggestion. We have replotted the figures as per this reviewer's suggestion and provided as Supplementary Fig. 18 and 19 in the revised manuscript.

=>In the DSC cooling curve (Supplementary Fig. 15) and the temperature dependence of α (relative degree of crystallization) (Supplementary Fig. 18), we found that the crystallization of the 1/PEO composite starts earlier than the pristine bulk PEOs. This resulted in the higher (~ 1 °C) crystallization temperature for the composite compared to the pristine ones (Supplementary Fig. 15). In the case of PEG2k and 1/PEG2k1/1, the PEO crystallization proceeded rapidly with making similar trend of the crystallization curves in all cooling rates (Supplementary Fig. 18a,c). On the other hand, in the crystallization curves for PEO4M and 1/PEG4M1/1, the trend of crystallization curve was largely different. The crystallization of 1/PEG4M1/1 was significantly retarded as the crystallization proceed

(Supplementary Fig. 18b,d) while the pristine **PEO4M** showed the normal crystallization trend that is similar to that of **PEG2k**. We ascribe this significant retardation of the PEO crystallization to the penetrating structure in which each PEO chain is partially trapped by **1** particles via topological constraints.

Supplementary Fig. 18.

Evolution of relative degree of crystallization, α , of **PEO2k** (dotted lines) and **1/PEO2k_{1/1}** (solid lines) (a,c), and **PEO4M** (dotted lines) **1/PEO4M_{1/1}** (solid lines) (b,d) at cooling rate of 0.4 °C/min (purple), 0.6 °C/min (blue), 1 °C/min (green), 2 °C/min (yellow), and 5 °C/min (orange) against temperature (a,b) and crystallization time (c,d).

=>The reorganized Friedman plots were given in Supplementary Fig. 19. The trends observed for the pristine **PEO2k** and the corresponding composite looks similar (Supplementary Fig. 19a,b) while the pronounced difference was revealed for **PEO4M** and its composite (Supplementary Fig. 19c,d). The change in the slope displayed in the figures corresponds to the change in the activation energy of crystallization at the respective degree of crystallization, α . This clearly represents the effect of penetration structure in polypseudoMOFaxane.

Supplementary Fig. 19.

a–d, The Friedman plots based on Eq. 5 for **PEO2k** (a), **1/PEO2k_{1/1}** (b), **PEO4M** (c), and **1/PEO4M_{1/1}** (d) at α of 10%, 30%, 50%, and 70%.

6) Any study on the melt crystallization kinetics and mechanism? To understand the crystallization process of 1/PEO2k_{1/1} and 1/PEO4M_{1/1}, Ozawa, and Avrami or Tobin plots for the crystallization kinetics should be performed. The support for the schematic plot in Fig 5b is rather weak at the moment.

⇒ We performed Avrami plot for **PEO2k**, **PEO4M**, and their composites with **1**. The results were given in Review-Only Figure 1 below. For **PEO2k**, there was no obvious difference in the slope that corresponds to the crystallization dimension (n index), between the pristine and composite state (Review-Only Figure 1a,b). On the other hand, **PEO4M** showed a change of the slope in the last half of the crystallinity process. In the pristine **PEO4M**, the slope change from $n = \sim 4$ (the first stage) to ~ 1 (the second stage), indicating the growing domain changes from spherite (3D) to rod (1D) (Review-Only Figure 1c,d and Review-Only Table 1). This two-stage crystallization process is commonly observed for PEO with relatively high molecular weight (e.g. *J. Polym. Res.* **2011**, *18*,

875–880). For **1/PEO4M_{1/1}** composite, similar two-stage crystallization process was observed, and the slope changed from $n = \sim 4$ (the first stage) to ~ 1.5 (the second stage) (Review-Only Figure 1c,d and Review-Only Table 1). The n index at the second stage was slightly larger than that of the pristine sample. Although the mechanism of this slight increase of n at the second stage is not clearly understood, this result supports the involvement of **1** microcrystals in the later stage of the PEO crystallization. In addition, the rate parameters, k' , of **1/PEO4M_{1/1}** were smaller than those of the pristine **PEO4M** at all given cooling rates (Review-Only Table 1), indicating the retardation of crystallization due to the presence of **1** crystals. This is consistent with the results obtained by the Friedman plot analysis (Fig. 6b). We note that the rate constants, k' , in the Review-only Table 1 are corrected with the consideration of the non-isothermal condition of the experiment as Jeziorny suggested (*Polymer* **1978**, *19*, 1142–1144):

$$\log k' = \frac{\log k}{\beta}$$

where k is the Avrami rate constant and β is cooling rate. Although these Avrami plots give such provisional insights in the crystallization process, we would like to make this discussion Review-Only as we consider that more systematic study is needed to unambiguously conclude the origin of observed changes in n index values.

Review-Only Fig. 1.

a–d, The Avrami plots **PEO2k** (a), **1/PEO2k_{1/1}** (b), **PEO4M** (c), and **1/PEO4M_{1/1}** (d) in the range

of relative degree crystallinity α between 1% and 98%.

Review-Only Table 1.

The Avrami parameters of **PEO4M** and **1/PEO4M_{1/1}** at various cooling rates.

Sample	Cooling rate (°C/min)	First stage		Second stage	
		n_1	$\log k_1'$	n_2	$\log k_2'$
PEO4M	0.4	3.48	-8.28	0.98	-1.54
	0.6	4.21	-6.41	0.83	-0.84
	1	3.38	-2.26	0.92	-0.32
	2	3.56	-0.79	0.83	-0.061
	5	4.36	-0.17	0.94	0.011
1/PEO4M_{1/1}	0.4	3.24	-9.88	1.42	-4.51
	0.6	3.59	-6.45	1.28	-2.36
	1	3.94	-3.44	1.56	-1.36
	2	4.75	-1.26	1.42	-0.39
	5	3.40	-0.033	1.50	-0.0072

7) With the additional work in above Points 5 and 6, the author can further elaborate the thermal degradation and crystallization of the polymer-MOF composite, together with Fig. 5a and 5b. All these will point to the merits of using polymer-MOF composite, where this is lacking in the manuscript at the moment.

=>As the response to above comment 5 and 6, we have included the following discussions based on the newly added figures (Supplementary Fig. 18 and 19) regarding the PEO crystallization in Section V.1 of the Supplementary Information.

(Supplementary Information)

(Page 19, line 22) “To facilitate the discussion of the PEO crystallization behavior, the data of Supplementary Fig. 16 and 17 were reorganized and replotted in Supplementary Fig. 18 and 19, respectively.

In the DSC cooling curves (Supplementary Fig. 15) and the temperature dependence of α (Supplementary Fig. 18), it was observed that the crystallization of the **1/PEO** composite starts earlier than that of the respective pristine PEOs. This resulted in the higher (~1 °C) crystallization temperature of the composites (Supplementary Fig. 15). In the case of **PEG2k** and **1/PEG2k_{1/1}**, the PEO crystallization proceeded rapidly with making similar trend of the crystallization curves in all cooling rates (Supplementary Fig. 18a,c). On the other hand, in the crystallization curves for **PEO4M** and **1/PEG4M_{1/1}**, the trend was largely different. The crystallization of **1/PEG4M_{1/1}** was significantly retarded as the crystallization proceeds (Supplementary Fig. 18b,d) while the pristine **PEO4M**

showed the normal crystallization trend that is similar to that of **PEG2k**. We ascribe this significant retardation of the PEO crystallization to the penetrating structure in which each PEO chain is partially trapped by **1** microcrystals via topological constraints.

In Supplementary Fig. 19, the trends observed for the pristine **PEO2k** and the corresponding composite looks similar (Supplementary Fig. 19a,b) while the pronounced difference was observed for **PEO4M** and its composite (Supplementary Fig. 19c,d). The change in the slope displayed in the figures corresponds to the change in the activation energy of crystallization at the respective degree of crystallization, α . This represents the effect of penetration structure in polypseudoMOFaxane (Fig. 6).”

- 8) The terminology used “MOFaxane” in which the author used the analogy to polyrotaxane, but polyrotaxane has end-caps at the end of the polymers. However, there’s no end caps for this polymer-MOF composite. Can the authors clearly define what MOFaxane mean? I’m not convinced to use this new terminology at the moment.

When I first read the word, MOFaxane, I had the first impression to relate to MXenes or MAXenes which are 2D structure; which clearly isn’t the case and the structure is completely different.

The authors are speculating the new polymer-MOF composite would have similar characteristics as that of polyrotaxanes. But there’s no solid evidence suggesting that the polymer-MOF composite has MOF sliding on the polymer chain, analogous to the cyclodextrin rings sliding on the axle chain for polyrotaxane (as on page 11 and 12 of the manuscript).

The author must think seriously in using the term MOFaxane, to prevent confusion and misrepresentation of the true property of this polymer-MOF composite.

=>We appreciate this important feedback. We consider that the terminology is very important as it is also pointed out by Reviewer 1. In the revised manuscript, we have added the definition of MOFaxane family with depicting the subclassified structural variations. Further, we revised Abstract and Introduction part with clearly mentioning that what we synthesized in this work is polypseudoMOFaxane, by definition, that has no end caps. Please refer to our response to the comment 1 from Reviewer 1. We believe that the newly added definition statements and revised Figure 1 are effective to avoid potential confusion of MOFaxane and other preexistent terms like MXene and MAXene.

=>We do not think that the presence of sliding motility is the prerequisite for rotaxane. The sliding activity is merely the matter of activation energy of translational diffusion of the rotary molecule on the axle (i.e., how easy the rotor can move on the axle). This dynamic character of rotaxane is dependent on many conditions such as temperature, solvent, as well as relative diameter of the rotor to the thickness of the axle (namely, so-called friction effect). Therefore, considering the definition of rotaxane, which focus on the molecular architecture, the mechanical property expected from the sliding motion is not always necessary. In the revised manuscript, we have calculated the adsorption

enthalpy of PEO in the MOF **1** to be 32 kJ/mol (per PEO repeating unit). According to the previous study on polyrotaxane formation (*Macromolecules* **1997**, *30*, 3685–3690), the binding energy of α -cyclodextrin (CD) and PEO chain is calculated to be ~ 30 kJ/mol (per PEO repeating unit). Therefore, the binding energy value for **1** and PEO chain is comparable to that of the traditional CD-PEO polyrotaxane system. However, in the present polyMOFaxane system, PEO chains are trapped with multiple interactions in the narrow pore with ~ 80 nm long, which makes the diffusion of PEO chain significantly slow. In our previous work, we reported that the effective diffusion constant of **PEO20k** in a MOF that has similar narrow channel with the diameter of 0.57 nm is 3.1×10^{-14} m²/s (40 °C) (*Chem. Sci.* **2021**, *12*, 12576–12586). This is 10^4 times slower than the diffusion constant of CD of the conventional polyrotaxane in solution, which has been experimentally determined to be 1.1×10^{-10} m²/s using quasi-elastic neutron scattering (QENS) measurements (30 °C) (*J. Am. Chem. Soc.* **2019**, *141*, 9655–9663). In the present polyMOFaxane system, we can provisionally consider that the *cp-to-op* deformation speed of **1** upon PEO insertion corresponds to the PEO diffusion rate in the pore. As can be seen in Figure 3b, the deformation takes some time ranging from minutes to hours depending on the PEO molecular weight. Therefore, we suppose that this slow chain diffusion leads to significantly slow relaxation time, making the sliding motion not to be manifested in the bulk mechanical properties of the polypseudoMOFaxane at an ambient temperature (25 °C). We also demonstrated in the previous work that MOFs with much larger pore diameter or higher temperature result in faster diffusion of polymer chains in the pore (*Chem. Sci.* **2021**, *12*, 12576–12586). From these considerations, we believe that further optimization of MOF structures and conditions would allow for more clearer observation of the anticipated mechanical properties arising from the characteristic sliding motion.

=>Please see our response to the comment 4 of Reviewer 1, in which the responses to this comment were also given.

9) The authors should update the list and make reference to other polymer threaded MOF publication: *J. Am. Chem. Soc.* 2014, *136*, 20, 7209–7212 and *ACS Energy Lett.* 2021, *6*, 11, 3769–3779

=>In response to this constructive feedback, we have cited following literatures including the suggested ones in the revised manuscript.

(References)

29. Kitao, T., Zhang, Y., Kitagawa, S., Wang, B. & Uemura, T. Hybridization of MOFs and polymers. *Chem. Soc. Rev.* **46**, 3108–3133 (2017).

30. Gao, L., Li, C.-Y. V., Chan, K.-Y. & Chen, Z.-N. Metal–Organic Framework Threaded with Aminated Polymer Formed in Situ for Fast and Reversible Ion Exchange. *J. Am. Chem. Soc.* **136**, 7209–7212 (2014).

31. Ho, C.-K. et al. Protonated Emeraldine Polyaniline Threaded MIL-101 as a Conductive High Surface Area Nanoporous Electrode. *ACS Energy Lett.* **6**, 3769–3779 (2021).

10) For mechanical property of the polymer-MOF composite, any results for 1/PEO2k1/1 and 1/PEO4M1/1? Should include for completion.

=>The mechanical tests of 1/PEO = 1/1 (wt/wt) composites could not be performed since the composites cannot be processed into robust free-standing film. Because of the high fraction of solid phase (50 wt%), they appear as powdery material with no ductility and flexibility. Due to this reason, we used large excess of PEO (1/10 and 1/20) for the preparation of free-standing film. Notably, even at 1/10 and 1/20 ratio, **PEO2k** did not give free-standing film because such short PEO cannot penetrate **1** microcrystals to form topological crosslinks. Additionally, due to another technical issue of film preparation, we could not measure mechanical properties of **PEO4M**-based composite. Since **PEO4M** gives highly viscous solution due to its extremely high molecular weight, it was technically difficult to obtain homogeneous solution in large scale. This issue prevented preparation of smooth film with uniform thickness. After the rigorous optimization of film preparation condition, we chose **PEO200k** and measured the mechanical properties of its composite film in the present work.

Reviewers' Comments:

Reviewer #1:

Remarks to the Author:

The raised concerns have been well addressed by authors using detailed data. I am now glad to recommend the publication of the manuscript as is.

Reviewer #2:

Remarks to the Author:

1) clarification on MOFaxane family in Fig. 1.

In this work, according to the author definition, this work are polypseudoMOFaxane.

Other forms of MOFaxane Fig. 1a-1c hasn't been fabricated, so it's sort of confusing to the readers.

The title is misleading at the moment. It's not MOFaxane, it should be revised accordingly to reflect the true essence of this work.

2) other revisions are satisfactory.

Point-to-Point Responses to Reviewer's Comments

For Reviewer 2:

1) clarification on MOFaxe family in Fig. 1. In this work, according to the author definition, this work are polypseudoMOFaxe. Other forms of MOFaxe Fig. 1a-1c hasn't been fabricated, so it's sort of confusing to the readers.

=>We are grateful for this important feedback. In Figure 1, we have added a term “**This work**” referring to polypseudoMOFaxe (d) as the compound fabricated in the present work. The corresponding statement has been also added to the caption of Figure 1 as follows.

(Revised Figure 1)

Fig. 1. Definitions and graphical representations of MOFaxe family. a, MOFaxe, b, pseudoMOFaxe, c, polyMOFaxe, and d, polypseudoMOFaxe. PseudoMOFaxe has no end cap on the polymer chains while MOFaxe has end caps by definition. PolyMOFaxe and polypseudoMOFaxe have network structures in which each polymer chain threads multiple MOF microcrystals, forming a topological network. In this work, polypseudoMOFaxe, d, was fabricated.

2) The title is misleading at the moment. It's not MOFaxe, it should be revised accordingly to reflect the true essence of this work.

=>We thank this reviewer for the useful comment on the title. We have revised the title in response to this comment. The new title is as follows.

(Revised Title)

“An approach to MOFaxes by threading ultralong polymers through metal–organic framework microcrystals”